# Discourse Sense Flows: Modelling the Rhetorical Style of Documents across Various Domains

**René Knaebel** and **Manfred Stede**

Applied Computational Linguistics
Department of Linguistics
University of Potsdam
Germany
{rene.knaebel,manfred.stede}@uni-potsdam.de

## Abstract

Recent research on shallow discourse parsing has given renewed attention to the role of discourse relation signals, in particular explicit connectives and so-called alternative lexicalizations. In our work, we first develop new models for extracting signals and classifying their senses, both for explicit connectives and alternative lexicalizations, based on the Penn Discourse Treebank v3 corpus. Thereafter, we apply these models to various raw corpora, and we introduce 'discourse sense flows', a new way of modeling the rhetorical style of a document by the linear order of coherence relations, as captured by the PDTB senses. The corpora span several genres and domains, and we undertake comparative analyses of the sense flows, as well as experiments on automatic genre/domain discrimination using discourse sense flow patterns as features. We find that n-gram patterns are indeed stronger predictors than simple sense (unigram) distributions.

## 1 Introduction

People write differently, and the style of writing highly depends on the author's intention. While texts from the news are largely informative, an argumentative essay, for example, is written to support a particular opinion. Understanding these underlying purposes and structures in a text—or why the text is arranged as it is—is a complex problem and requires giving more attention to differences between genres (e.g., news versus review) and domains (e.g., economics versus science).

Previous research has examined some genre differences within a single newspaper corpus (Webber, 2009), and more recent work has shown the impact of joining diverse genres in training data for discourse parsing (Liu and Zeldes, 2023). Similarly, Stepanov and Riccardi (2014) demonstrated that it is difficult to generalize "discourse knowledge" from one domain to others, e.g., from news to biomedical text. However, there is so far not much work on genre/domain differentiation on the basis of discourse features for large amounts of text documents.

We address this task from the perspective of discourse relations as operationalized in the Penn Discourse Treebank (PDTB) (Prasad et al., 2018). It represents discourse relations in the form of two text spans (*arguments:* Arg1 and Arg2) that are associated with a *sense*. These senses are organized in a three-level hierarchy and most research focuses on the first two levels, *coarse* and *fine* (see Table 2).

In our work, we for now rely only on the discourse signals and do not consider the arguments. Here is an example from the PDTB corpus with three signals:

(1) "We had been soliciting opinions on it [**long before**]₁ Mr. Lawson's resignation, [**and**]₂ offer some of the collection for the benefit of his successor and one-time deputy, John Major. [**To begin with,**]₃ we should note that in contrast to the U.S. deficit, Britain has been running largish budget surpluses." (*pdtb3_01793*)

The PDTB distinguishes two groups of signaled relations: **Explicit** relations are signaled by a closed set of discourse connectives, e.g. *because*, *and*, *if-then*, and *before*. **Alternative lexicalizations** (AltLex) use phrases other than connectives, e.g. *to begin with*, *for that reason*, and *it all adds up to*. While the first two signals in ex. (1) are explicit signals (and their senses are *Temporal.Asynchronous* and *Expansion.Conjunction*, respectively), the last one represents an alternative lexicalization with sense *Expansion.Instantiation*.

In total, the PDTB (version 3) contains 25,878 signaled relations, 94% of which are of type 'explicit', and the rest are AltLex phrases. Importantly, the PDTB also contains 'implicit' relations that are not lexically signaled at all; for our present purposes, we do not use them.

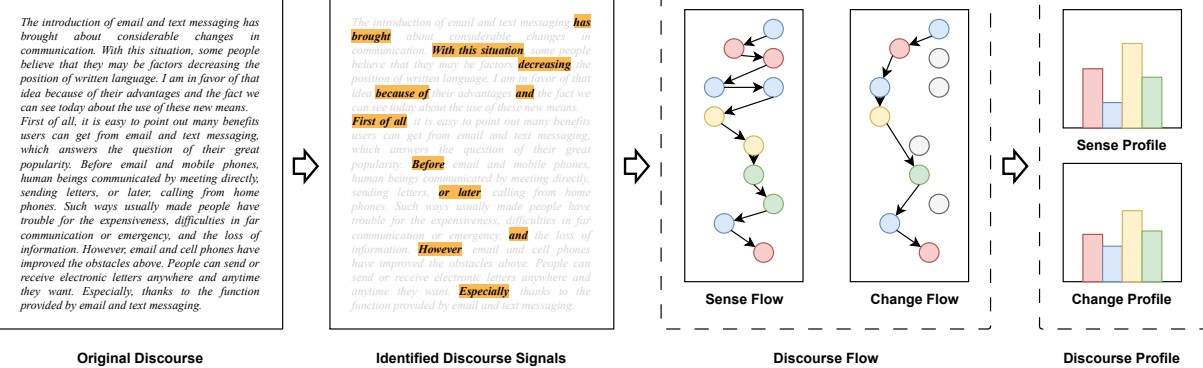

Figure 1: Overview of the signal extraction process, from plain text via the modeling of sense flows to the abstraction of discourse 'profiles'. In this figure, we illustrate senses by only four classes.

In this paper, we develop the first models for identifying explicit and AltLex signals and their senses from the recent version 3 of the PDTB. Then, we use the models on corpora of raw texts from different domains and genres, and we demonstrate that automatically produced signal/sense information can be used to differentiate those domains. Figure 1 illustrates the steps from raw documents to 'sense profiles' that we regard as models of the document's rhetorical style. They are operationalized by the distributions of n-gram patterns extracted from the document's sense flow.

**Contributions** In short, we (i) develop novel models for classifying senses of discourse signals (connectives and AltLex phrases) based on the PDTB v3 corpus. We (ii) introduce a new concept called *discourse sense flows*, extract patterns of these flows, and provide statistical analysis across nine corpora from various genres and domains. Based on this concept, we (iii) examine statistical models' abilities to distinguish the rhetorical style of documents stemming from those genres/domains, in terms of sense n-grams.

We summarize the related work (§2) and the corpora we use (§3). Then, we describe our signal extraction models (§4). Next, we introduce sense flows and study their predictions on texts from other domains (§5).

## 2 Related Work

**Signal Identification** Traditionally, explicit connectives were identified, first, by accessing precompiled connective lexicons and matching possible candidates, before a statistical model learns to disambiguate these candidates for their 'discourse' or 'sentential' use. Using a maximum entropy model, early work by Pitler and Nenkova (2009) disambiguated explicit connectives and their associated coarse sense based on syntactic features from gold constituency trees such as categories of the connective, its parent and siblings, as well as pairwise interaction features. Later, the feature set was elaborated, e.g., with lexico-syntactic features and root-to-connective category paths (Lin et al., 2014), and additional syntactic context information (Wang and Lan, 2015). Recent work integrated contextualized word embeddings for identification and sense classification (Knaebel and Stede, 2020).

Other work eliminated the dependency on precompiled lexicons by incorporating transformer-based sequence labeling approaches. Kurfalı (2020) used BERT to successfully predict single and consecutive multi-word connectives. With respect to the CoNLL shared task's exact evaluation scheme (Xue et al., 2016), he achieves a 96.77 F1 score for connective identification in the PDTB (v2). However, a simplification has been introduced that replaces a full connective by its corresponding head, e.g. *after 10 minutes* is mapped to *after*, which makes the task significantly easier.

Recognizing AltLex signals in the PDTB corpus is challenging due to the higher complexity of the phrases when compared to explicit signals, and due to the small number of training examples. Prasad et al. (2010) analyzed regularities by defining groups of phrases in the older version of the PDTB. In shallow discourse parsing, The identification of AltLex phrases was generally neglected, or the AltLex instances were simply merged with the implicit relations. Therefore, in the context of the above-mentioned CoNLL shared task no dedicated previous results for AltLex sense classification were available. A first approach to predicting

the presence of AltLex phrases based on PDTB v3 was presented by Knaebel and Stede (2022). A shared task on multi-formalism discourse parsing (Zeldes et al., 2021) formed a new notion of signal identification by combining explicit and AltLex relation signals, but it considered discontinuous signals (e.g., *if .. then*) as separate relations. The task is thus different and results cannot be easily compared. Best performing systems in the connective identification track achieved scores of 92.02 F1 (Gessler et al., 2021) and 91.15 F1 (Bakshi and Sharma, 2021) respectively, both using a combination of LLM and CRF for sentence labeling.

**Rhetorical Style**   Pitler et al. (2008) show that certain adjacent discourse relations are not independent of each other, thus motivating the identification of certain local sense patterns. Lin et al. (2011) use their previously developed PDTB parser to identify discourse relations and their senses (only coarse-grained), and they introduce the concept of *discourse role matrix* to model coherence in texts. The focus is on handling sentence permutation and on applications such as summarization and argumentative zoning. Biran and McKeown (2015) develop an n-gram model on PDTB senses that is used for planning a story. Their work shows performance differences depending on the n-gram model's training domain. Braud and Søgaard (2017) detect scientific fraud, with features including token n-grams and explicit connectives with their associated coarse senses. Ji and Smith (2017) developed a recursive neural model for document classification according to the rhetorical structure (RST) of the document.

Wachsmuth et al. (2015) introduce *sentiment flows* and generalize from local sentiments to the global sentiment of a review; they proposed using sentiment flow patterns for classification purposes. Building on this idea, Al-Khatib et al. (2017) adapt change flows to argumentation mining for identifying patterns of evidence types and their correlations to the topic of a document. In contrast to these two works that consider flows as a whole, we instead extract n-grams from discourse sense flows, for identifying patterns in rhetorical styles.

## 3   Data

This work examines the different rhetorical styles found in documents and investigates whether documents can be associated with their corresponding

| Domain | Genre | #Docs | #EX | #AL |
|---|---|---|---|---|
| ACL | ABSTRACT | 9,983 | 30,309 | 2,833 |
| MED | ABSTRACT | 10,000 | 32,498 | 3,607 |
| AES | ESSAY | 1,562 | 31,456 | 1,077 |
| PEC | ESSAY | 402 | 6,395 | 440 |
| NYT | NEWS | 43,540 | 49,253 | 1,605 |
| BBC | NEWS | 2,225 | 25,116 | 1,211 |
| WSJ | NEWS | 2,161 | 24,218 | 1,536 |
| TED | SPEECH | 2,459 | 172,526 | 5,825 |
| UN | SPEECH | 5,000 | 36,545 | 2,608 |

Table 1: Corpus Statistics Overview: All relation counts are approximations based on automatic predictions.

corpus, in the following referred to as *domain*, and also whether different domains have certain patterns in common. Therefore, we prepare additional corpora, summarized in Table 1, to complement the data in the PDTB. We group domains into four specific *genre* classes: ABSTRACT (scientific abstracts from two domains), ESSAY (argumentative learner essays), NEWS (two additional news wire agencies with various topics), and SPEECH (casual and political speech transcripts). We expect these genres to differ in their style of writing due to their varying goals (e.g. informative, persuasive, entertaining) and target audiences, respectively. As some of the datasets were very large, we scaled them to a computationally feasible size as described below.

The *Penn Discourse Treebank* (PDTB) (Prasad et al., 2018) consists of 2,499 stories from a three-year Wall Street Journal (WSJ) collection, and thus belongs to the genre of News Articles. We use this corpus for training as it is by far the largest available source for discourse relation annotations. While the previous version contains around 43K annotations with 18.4K explicit and 16K implicit relations, the recent version 3 introduces a new annotation scheme with a revised sense hierarchy as well as 13K additional relations.

In the following, we briefly summarize our additional raw corpora grouped according to their genre.

**News**   The *New York Times Annotated Corpus* (NYT)[1] contains about 1.8 million documents published by the New York Times between 1987 and 2007. We sample 500 documents per month from the years 2000–2007, but remove some of the articles due to peripheral contents, e.g., business

---

[1] https://doi.org/10.35111/77ba-9x74

market overviews, and thus focus on the more recent part of the dataset. The *BBC News Articles* (BBC) (Greene and Cunningham, 2006) consists of 2225 documents from the BBC news website corresponding to stories in five topical areas (business, entertainment, politics, sport, tech) from 2004–2005.

**Essays** The *Automated Essay Scoring* corpus (AES)[2] was created for a shared task by the Hewlett Foundation in 2012. This corpus consists of manually-scored student essays for different prompts and grade levels. For our study, we selected only the 1,500 essays of the persuasive task written by 10th-grade students. The *UKP Argument Annotated Essays* corpus (PEC) (Stab and Gurevych, 2017) consists of 402 persuasive essays written by language learners (crawled from an essay website).

**Speeches** The *TED Conferences Transcripts* (TED)[3] is a collection of English transcripts crawled from the TED site. It contains about 2500 talks with publication years ranging from 2006 to 2017. We add artificial paragraph breaks whenever audience information such as applause was noted. The *UN Security Council Debates* corpus (UN) (Schoenfeld et al., 2019) is based on 5,748 meeting protocols dated between 1995 and 2020. The protocols are split into distinct speeches and result in 82,165 individual documents. Documents from 2018–2020 have been removed due to some conversion errors in the dataset. We randomly select 5000 documents from the corpus.

**Abstracts** The *ACL Anthology Reference Corpus* (ACL) (Bird et al., 2008) is a continuously updated image of the publicly available ACL Anthology that collects conference and journal papers in the areas of natural language processing and computational linguistics. We sample 10,000 English abstracts from the years 2000–2019. Another publicly available source of scientific literature is the *PubMed* (MED)[4] search engine that provides access to the MEDLINE database. Here, we also sample 10,000 abstracts, but due to the missing publication meta information, we do not further restrict the data.

---

[2] https://www.kaggle.com/competitions/asap-aes
[3] https://www.ted.com/
[4] https://pubmed.ncbi.nlm.nih.gov

| Level-1 | Level-2 | #EX | #AL |
|---|---|---|---|
| TEMPORAL | SYNCHRONOUS | 1,947 | 29 |
| | ASYNCHRONOUS | 2,111 | 130 |
| CONTINGENCY | CAUSE | 2,395 | 915 |
| | CONDITION | 1,686 | 32 |
| | PURPOSE | 381 | 34 |
| COMPARISON | CONCESSION | 4,868 | 31 |
| | CONTRAST | 1,353 | 41 |
| | SIMILARITY | 109 | 12 |
| EXPANSION | CONJUNCTION | 8,799 | 138 |
| | DISJUNCTION | 305 | 0 |
| | EQUIVALENCE | 27 | 10 |
| | EXCEPTION | 36 | 3 |
| | INSTANTIATION | 325 | 61 |
| | LEVELOFDETAIL | 271 | 62 |
| | MANNER | 543 | 3 |
| | SUBSTITUTION | 219 | 29 |

Table 2: Simplified Sense Hierarchy: counts of explicit relations (EX) and alternative lexicalizations (AL).

## 4 Identification of Signaled Discourse Relations

For our study, we train models for both tasks, identification of discourse signals and classifying their sense, separately. Similar to Kurfalı and Östling (2021), for better context information we process full paragraphs instead of individual sentences. For signal labeling, we use a pre-trained large language model (LLM) and fine-tune the base model combined with a linear transformation and a conditional random field (CRF) (Lafferty et al., 2001) as output layer. For properly encoding discourse signals, we use BIOES encoding. This has the advantage that we can encode all types of discourse signals without loss: discontinuous signals such as *not only [. . . ] but also [. . . ]* and also signals that are nested within each other. We further add *start* and *end* tags for technical reasons related to the CRF.

For relation sense identification, we follow most previous research and focus on the second level rather than on the coarse senses. Due to the small amount of samples available for some classes, we ignore those that are distinguished by the features negation, speech act, and belief. Table 2 shows our proposed simplified sense hierarchy used throughout our work. For comparison purposes, however, we will also show results on the original fine-grained (second-level) senses. We use the same LLM for signal labeling, but pre-compute token embeddings per paragraph and use them as fixed

| Model | Partial-Match | Exact-Match |
|---|---|---|
| Explicit (Linear) | 95.74 | 95.69 |
| Explicit (CRF) | 95.96 | 95.91 |
| AltLex (Linear) | 58.47 | 56.94 |
| AltLex (CRF) | 60.44 | 58.16 |

Table 3: Labeling performances: macro F1 for partial and exact matches, comparison of Explicit and AltLex models with linear and CRF prediction layers.

| | Model | Eval | Coarse | | Fine | |
|---|---|---|---|---|---|---|
| | | | Acc | F1 | Acc | F1 |
| **Simplified** | altlex | altlex | 91.22 | 81.11 | 86.91 | 60.73 |
| | | explicit | 55.87 | 49.76 | 38.78 | 25.23 |
| | explicit | altlex | 77.73 | 62.36 | 71.65 | 49.49 |
| | | explicit | 92.20 | 84.86 | 87.09 | 78.14 |
| | both | altlex | 90.14 | 78.60 | 86.70 | 62.62 |
| | | explicit | 92.02 | 86.38 | 86.90 | 78.17 |
| **Original** | altlex | altlex | 91.73 | 81.23 | 86.12 | 56.42 |
| | | explicit | 56.22 | 47.63 | 38.48 | 20.44 |
| | explicit | altlex | 77.58 | 66.87 | 69.25 | 44.08 |
| | | explicit | 92.20 | 86.62 | 86.72 | 68.38 |
| | both | altlex | 90.32 | 80.91 | 87.71 | 61.85 |
| | | explicit | 92.00 | 86.42 | 86.57 | 66.81 |

Table 4: Sense classification performances (accuracy and macro f1 score), comparison of models trained on standard and simplified coarse and fine senses, either on one of the signals or on both.

inputs for our sense classifiers. Given a discourse signal, we aggregate all its token embeddings by mean and max pooling to get the same input size due to varying signal lengths. Furthermore, we concatenate the respective surrounding embedding, left and right, to the model input, following (Knaebel and Stede, 2020). We jointly train coarse and fine senses, as it has been shown beneficial for overall learning. (Long and Webber, 2022)

## 4.1 Quality of Extracted Discourse Signals

As suggested by Shi and Demberg (2017), we perform cross-validation of the results of our two models by repeating the training 10 times with random initialization and present mean scores with standard deviations. We split 10% of the PDTB for testing purposes and, further, split 10% of the remaining data for validation during training. Detailed hyper-parameters are shown in Appendix A. We train our labeling models in two stages: First, we optimize the linear classification layer that later provides emission scores to the CRF using cross-entropy loss. Second, we optimize the negative log-likelihood given the emission scores for a sequence of output tags using the forward algorithm. Final outputs for evaluation are computed using the Viterbi algorithm.

For the evaluation of discourse signal extraction, we use partial matching (Xue et al., 2016) and define the matching overlap based on the F1 score of two connecting phrases. *Partial-Match* and *Exact-Match* refer to 70% and 90% F1 matching thresholds, respectively. For example, when our model recognizes two of three words of the signal *to begin with* correctly, resulting in 0.66 recall, 1.0 precision, and thus 0.83 F1, this signal would count as partially matched but not exactly. All experiments run on the same test set per iteration, with varying training and validation splits, 10 repetitions each.

In Table 3, we compare the base architecture (the linear classifier layer), with the extra CRF final layer. For both prediction tasks, using a CRF is beneficial: while for the explicit connective identification, the performance gains are rather small and the overall improvement is mainly rooted in better recall values, we notice a higher increase in performance for AltLex signals of about 2% F1 score. We attribute this result to the different average lengths of the signals, as explicit signals tend to be shorter.

Comparison with previous state-of-the-art (Gessler et al., 2021) is not trivial, due to slightly different problem settings. The DISRPT shared task divided discontinuous signals into separate ones, thus simplifying the task; however, they use a stricter evaluation threshold. Combining our separate results for Exact-Match on explicit and AltLex signals by their relative occurrence within the PDTB, we arrive at a competitive result of 93.65 F1 score.

For evaluating the raw sense classification (see Table 4), we compare both sense hierarchies, the official PDTB version and our simplified version. We train sense models on gold signals of either one of the signal types, AltLex or Explicit. Further, we train a third model on the joint set of signals and thus investigate whether one model has advantages from learning from the other types of signals. Comparing both sense hierarchies, the coarse and fine sense predictions are rather mixed. For the original hierarchy, the coarse senses have better performance, while we have the opposite in the

| Both Model | Partial-Match | Exact-Match |
|---|---|---|
| AltLex (simple) | 72.28 | 69.09 |
| AltLex (std) | 71.76 | 68.66 |
| Explicit (simple) | 94.85 | 94.76 |
| Explicit (std) | 94.25 | 94.16 |

Table 5: Sense Classification performances with error propagation: macro F1 average for partial and exact matches, comparison of Explicit and AltLex model for standard (std) and simplified (simple) senses.

case of our simplified hierarchy. We hypothesize the model to focus on the easier task (classifying 4 types) rather than the more complex fine-sense classification, thus resulting in better coarse-sense scores. However, as we are mainly interested in fine senses, seeing coarse senses more as an auxiliary task, we refrain from further investigations.

For fine senses, performance increased in our simplified hierarchy compared to the original one. This can indicate that the ignored features (implicit beliefs, speech acts, negation), are too hard to learn for the model due to the missing argument spans. Training both signal types jointly seems to have mainly negative effects, as there is neither an advantage over specialized models nor the simplified sense hierarchy leading to better results.

Table 5 summarizes the results of our sense classification model jointly trained on both signal types. The results are based on extracted signals and thus are similar to the actual parsing performance from previous shared tasks, for example. As in raw sense classification experiments, the standard hierarchy results are outperformed by the simplified version. For our subsequent experiments, we continue with this pipeline and the simplified sense hierarchy.

At the CoNLL Shared Task (Xue et al., 2016), the best system for partial matching of explicit relations (Wang and Lan, 2016) achieved 69.21 F1, while we report simplified scores of 94.85 F1. Simplified senses seem much closer to the Shared Task senses, although they still have a smaller number, however, we also have to note that this result is highly influenced by the results on argument extraction. Comparing our 78.14 F1 for explicit sense classification, (Kido and Aizawa, 2016) achieves 90.22 F1, but with gold arguments and fewer senses to classify.

## 4.2 Cross-Domain Discourse Signals

For cross-domain experiments, we train the same models as presented in Section 4, but omit the additional test split and thus have more training data available, which is especially helpful for alternative lexicalizations. Further, we train ensembles of size three for each of the tasks, separate explicit and AltLex signal labeling as well as joint sense classification, and combine their outputs by majority voting and mean of output distributions, respectively. In this way, the combination of all models had access to all available training data, and the overall prediction results should improve. [5]

**Explicit Connectives**   For comparison with the originally annotated connectives, we utilize the connective heads from the PDTB and we merge our extracted connectives into this list. Especially for temporal connective heads such as *after*, *before*, and *later*, often combined with numerical values and duration (e.g. 10 minutes, three days), we easily find a high number of variants recognized by our model. Also, adverbial modifications in combination with *because* produce more variants that are not recognized in training data, e.g. *probably*, *mostly*, and *precisely*.

Within the set of extracted connectives that do not match the connective heads list, we found some adjective/adverb ordinals, e.g. *first*, *secondly*, etc., that would rather belong to the group of AltLex signals. Other adverbs such as *actually*, *essentially*, *fortunately*, and *obviously*, also occur occasionally, which seem to not have any real discourse usage. Interestingly, we notice that misspellings of connectives, e.g., *besause*, *beacause*, and *becasue*, were identified as discourse signals.

The complete automatically compiled list, where connective heads are associated with their modified variants, is presented in Appendix D.

**Alternative Lexicalizations**   We identified 41,308 instances (9,359 types) of alternative lexicalizations in all corpora. AltLex signals from the official PDTB have an average length of 2.35 ($\pm$ 1.68), while our predictions' averages range from 1.88–2.72, though with a lower standard deviation. Interestingly, signals extracted from ESSAY and SPEECH are longer on average compared to the original PDTB.

Most of the unseen AltLex phrases belong to the

---

[5] Our models are available under https://github.com/rknaebel/emnlp-2023-discourse-signal-flows.

group of causal signals where we identified 1268 variants outside the PDTB. For signals associated with DISJUNCTION, EQUIVALENCE, EXCEPTION, MANNER, and SUBSTITUTION, all belonging to the Expansion class, we could not identify new instances. For a more complete list of senses and their associated AltLex phrases see Appendix E.

**Jointly Classified Discourse Senses**  Table 6 summarizes the relative frequencies of individual senses per corpus ordered by the overall usage. As expected, the most frequent relation by far is the CONJUNCTION relation, which occurs particularly often in SPEECH, 50.94% and 64.18%, and ABSTRACT genres, 58.99% and 57.54%, respectively. CONTINGENCY relations (CAUSE, CONDITION, PURPOSE) are used above average in the genres ESSAY and ABSTRACT, not surprisingly, as these genres are rather argumentative in nature. We notice that AES is slightly below average, possibly because of but errors due to incorrect spellings (learner language). Also, the higher frequency of PURPOSE and MANNER in the UNSC domain allows conclusions about the way politicians argue in their debates. For NEWS, we recognize very similar proportions of CONCESSION and the temporal relations SYNCHRONOUS and ASYNCHRONOUS. The latter represents the main elements of a news article: chronologically describing the events that happened.

# 5 Modeling Rhetorical Style by the Flow of Discourse Signal Senses

## 5.1 Discourse Sense Flows

Approaches that have been suggested for modeling the "rhetorical style" of text documents include trees formed by coherence relations, as for instance in Rhetorical Structure Theory (Mann and Thompson, 1988), and a "move" or "zone" analysis, where the text is modeled as a sequence of contiguous functional units (Teufel and Moens, 2002). For large-scale automatic analysis, however, RST parsing does so far not robustly yield good results across domains; and the "zone" approach deliberately uses genre-specific labels.

Alternatively, we here propose to model the global rhetorical style of a document as a linear sequence of discourse relations. We apply the signal identification models from the previous section and thus limit our model, for now, to *signaled* relations and thus speak of signal "sense flows"; implicit

coherence relations are left for future work.

In order to impose a linear order on the PDTB relations in a text, we anchor every relation at the position of the first token of its signal, which yields a sequence of senses. In addition, following Wachsmuth et al. (2015), we map a text's sense flow also to a *change flow* by collapsing contiguous instances of the same sense into a single one (thus a change flow thus does not contain any bigram of identical senses). Notice that for trigrams or larger $n$-grams, this elimination can result in new flow patterns that have not been observed before the elimination. We add special tags to the beginning and the end of a sense flow so that these positions are explicitly represented in n-gram models. The resulting n-grams for the text constitute the features for our "style" classifier, introduced in the next section. For illustration, we provide a sample of bigrams extracted from our corpora in Table 7; the frequencies demonstrate some noticeable differences between the different domains. Longer lists of the 50 most frequent sense flows and sense change flows are given in Appendix B.

## 5.2 Discourse Senses for Text Classification

Given the differences in proportions of sense n-gram patterns, we hypothesize that these patterns are useful for distinguishing domains from each other and that patterns are indeed stronger features than models that simply contain distributions of the single senses. We test this by comparing the performance of linear models trained on varying n-gram sizes.

**Model**  We process our data and sample at most 804 documents[6] per domain per run to reduce effects by different corpus sizes. This allows us to omit the linear model's intercept (bias) in order to focus on the combination of features exclusively. We split 20% of the data for evaluation purposes before we extract n-gram patterns ($n \in \{1 \dots 4\}$) from the sense flows of all texts. Then we use their TF/IDF-weighted frequencies for training maximum entropy (logistic regression) models because of their simplicity and interpretability. Besides extracting features from raw discourse sense flows, we also compare models trained on change flows (as described above).

**Results**  Our goal is to exploit differences in rhetorical style for distinguishing documents from

---

[6]This is twice as many documents as are available for PEC, where only the available 402 documents could be used.

| Discourse Sense | Mean | AES | PEC | ACL | MED | BBC | NYT | WSJ | TED | UN |
|---|---|---|---|---|---|---|---|---|---|---|
| CONJUNCTION | 48.914 | 34.018 | 40.173 | 58.993 | 57.545 | 45.623 | 45.633 | 43.103 | 50.945 | 64.189 |
| CONCESSION | 14.380 | 12.967 | 12.257 | 12.260 | 11.035 | 21.383 | 19.229 | 20.132 | 11.235 | 8.919 |
| CAUSE | 14.148 | 17.622 | 21.006 | 14.233 | 16.025 | 9.044 | 9.082 | 11.695 | 19.138 | 9.490 |
| CONDITION | 6.628 | 20.951 | 7.143 | 1.024 | 1.934 | 6.342 | 5.676 | 6.667 | 6.429 | 3.489 |
| ASYNCHRONOUS | 5.722 | 2.955 | 3.064 | 4.500 | 4.135 | 9.972 | 10.110 | 8.855 | 5.232 | 2.679 |
| SYNCHRONOUS | 2.945 | 2.506 | 2.346 | 1.345 | 0.965 | 3.459 | 5.626 | 3.467 | 3.474 | 3.315 |
| PURPOSE | 1.564 | 1.342 | 1.902 | 2.043 | 1.350 | 0.807 | 0.566 | 0.658 | 0.857 | 4.555 |
| DISJUNCTION | 1.479 | 4.297 | 1.775 | 1.081 | 0.965 | 0.882 | 1.333 | 1.248 | 1.049 | 0.683 |
| INSTANTIATION | 1.340 | 1.301 | 6.065 | 0.725 | 0.195 | 0.401 | 0.812 | 1.504 | 0.518 | 0.543 |
| CONTRAST | 1.273 | 0.790 | 2.240 | 1.112 | 4.894 | 0.486 | 0.451 | 1.055 | 0.227 | 0.202 |
| SUBSTITUTION | 0.848 | 0.985 | 1.395 | 0.866 | 0.221 | 0.943 | 0.889 | 0.987 | 0.598 | 0.748 |
| MANNER | 0.476 | 0.137 | 0.528 | 1.520 | 0.515 | 0.266 | 0.261 | 0.235 | 0.119 | 0.707 |
| LEVELOFDETAIL | 0.150 | 0.000 | 0.042 | 0.281 | 0.198 | 0.281 | 0.072 | 0.183 | 0.023 | 0.267 |
| SIMILARITY | 0.129 | 0.129 | 0.063 | 0.018 | 0.023 | 0.110 | 0.252 | 0.198 | 0.154 | 0.212 |
| EXCEPTION | 0.003 | 0.000 | 0.000 | 0.000 | 0.000 | 0.000 | 0.007 | 0.016 | 0.002 | 0.000 |
| EQUIVALENCE | 0.000 | 0.000 | 0.000 | 0.000 | 0.000 | 0.000 | 0.000 | 0.000 | 0.001 | 0.000 |
| # Documents | 59,545 | 1,562 | 402 | 1,754 | 8,390 | 2,198 | 38,755 | 1,820 | 2,455 | 2,209 |
| # Relations | 825,581 | 24,061 | 4,732 | 22,757 | 26,216 | 19,946 | 540,406 | 19,154 | 139,047 | 29,262 |
| Relations per Doc | 13.86 | 15.4 | 11.77 | 12.97 | 3.12 | 9.07 | 13.94 | 10.52 | 56.64 | 13.25 |

Table 6: Distribution of discourse senses on individual corpora, relative frequency in percent, entries sorted by the macro average per relation. Absolute counts are shown at the bottom, as well as their sum.

| | Flow Pattern | Mean | AES | PEC | ACL | MED | BBC | NYT | WSJ | TED | UN |
|---|---|---|---|---|---|---|---|---|---|---|---|
| 7 | Conjunction-<pad> | 2.37 | 1.26 | 2.38 | 5.18 | 3.82 | 2.50 | 1.03 | 2.25 | 0.83 | 2.12 |
| 40 | Cause-Condition | 0.46 | 2.01 | 0.79 | 0.07 | 0.11 | 0.18 | 0.14 | 0.17 | 0.49 | 0.15 |
| 44 | Condition-Cause | 0.44 | 2.01 | 0.79 | 0.07 | 0.11 | 0.09 | 0.13 | 0.17 | 0.42 | 0.15 |
| 46 | Concession-Asynchronous | 0.43 | 0.25 | 0.00 | 0.15 | 0.22 | 0.98 | 0.66 | 1.21 | 0.28 | 0.08 |
| 47 | Concession-Synchronous | 0.42 | 0.25 | 0.00 | 0.30 | 0.28 | 0.71 | 0.67 | 1.04 | 0.35 | 0.15 |
| 48 | Asynchronous-Concession | 0.41 | 0.13 | 0.00 | 0.15 | 0.28 | 1.16 | 0.67 | 1.04 | 0.21 | 0.08 |
| 49 | Synchronous-Concession | 0.41 | 0.25 | 0.00 | 0.22 | 0.22 | 0.71 | 0.65 | 1.04 | 0.35 | 0.23 |

Table 7: Selected discourse sense flow patterns of n-gram size 2 with relative frequency per corpus.

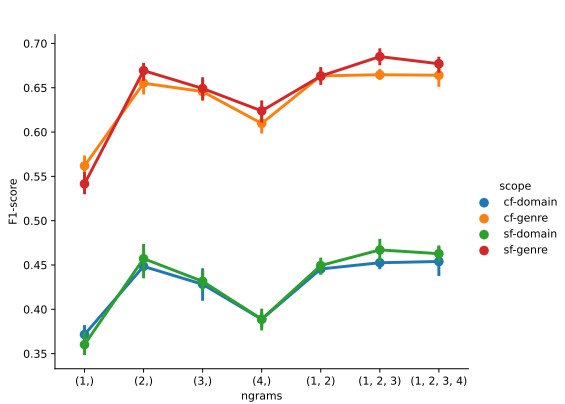

Figure 2: Pattern-based domain and genre classification. "Scope" indicates combination of flow type (sense flows (sf), change flows (cf)), and domain/genre distinction.

the various domains and their associated genres that we summarized in Section 3. A random stratified baseline trained on our sampled data achieves 0.11 macro F1 (±0.006) for the domain classification and 0.25 macro F1 (±0.007) for genre classification, respectively, which is clearly outperformed by our feature-based approach. Figure 2 shows our results for different n-grams. Larger n-gram sizes outperform single senses (unigrams). Furthermore, it turns out that bi-grams show the best single performance, while a combination of 1, 2, and 3-grams achieves the best overall performance. In our experiments, we noticed a decrease in performance whenever 4-grams were involved as features. In this regard, we hypothesize that a weak point of our modeling is the missing implicit relations, which account for half of the annotations. When linearizing the discourse senses, we have certain (implicit relation) gaps in the flow. With longer n-grams, the chance increases that extracted patterns contain missing discourse information. These patterns seem rather misleading for the classification model.

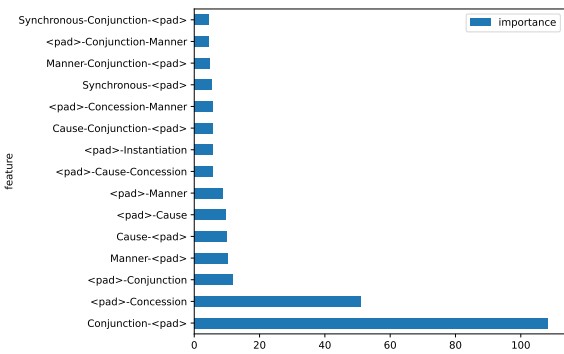

Figure 3: Feature importance of domain ACL based on the weights of linear model.

As for comparing sense flows to change flows, there is no significant difference for domain classification, while for genre classification the sense flows often outperform change flows.

As an example of feature importance derived from the weights learned by the statistical model, Figure 3 shows the 15 most important features for the domain of ACL abstracts. Besides the importance of causal relations, the model also learns that common introductions for ACL abstracts are CONCESSION, MANNER, and INSTANTIATION. Appendix C gives a complete view of the remaining classes and their most important features.

## 6    Conclusions

We introduced the first models for identifying both explicit and AltLex signals and their senses in the PDTB v3 corpus. The resulting system was then used on unseen corpora from different domains, and we propose the notion of discourse sense flows to model the rhetorical style of documents. Reducing complete flows to n-grams of different sizes enables us to discriminate certain genres and domains with a classifier that uses merely these automatically generated n-grams as features.

Our work is currently limited to discourse relations that contain lexical signals, e.g., explicit connectives and alternative lexicalizations. However, given a classifier for implicit relations (a task that received much attention in the literature in recent years), those relations can be integrated, and we assume this can lead to more explanatory sense flows and thereby to improved genre/domain classification performance.

In addition, we anticipate further applications of

discourse sense flows, for example in essay scoring or in text-level sentiment analysis.

Finally, assuming the availability of robust RST-style parsing, adapting the extraction of discourse sense flows to tree structures might be useful, either by linearizing the tree or extracting more complex tree patterns than sense n-grams.

## Ethics

We do not see any critical aspects of our work with respect to ethical considerations.

However, we want to highlight that our statistical models are trained only on English news-wire articles from the Wall Street Journal published around 1990. We are aware, that this highly influences the predictions made on other domains' corpora. Unfortunately, other resources for shallow discourse annotations are very limited.

## Limitations

Studying only the signaled relations provides a first but limited view of a document's rhetorical style. We think that adding the remaining relation classes of the theoretical framework will give much more valuable insights, also reflected by better prediction scores and better separations.

The model for signal extraction was trained and evaluated entirely on English data. However, due to the usage of a neural language model for encoding paragraphs, adapting the model to a different language is straightforward.

## Acknowledgements

This research has been supported by the German Research Foundation (DFG) with grant number 455911521, project "LARGA" in SPP "RATIO". We would like to thank the anonymous reviewers for their valuable feedback.

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

## A  Hyper Parameter Settings

All models are implemented in PyTorch with additional help from the Transformers library. Throughout our experiments, we use *roberta-base* architecture for encoding paragraphs. In the following, we summarize the settings for both model types that are finally used to extract signaling phrases and classify their associated sense.

**Signal Labeling**  The signal labeling model uses a batch size of 16. We start by optimizing the linear classifier and use Adam with a learning rate of $5e-5$ for the language model and $1e-3$ for the linear classification layer. During iterations, the gradient norm is clipped to 5. The model is trained for 10 epochs, early stopping if there is no improvement for 3 epochs. As recommended by fine-tuning RoBERTa, we use a linear scheduler for the learning rate and 50 warmup steps. We partially fine-tune the language model, in particular, we freeze the first half of the layers and only fine-tune the second half. Also, we use a dropout of 0.3 right after the language model and before the linear classification layer. For measuring performance, we chose the macro F1 score. After the training has stopped, we continue with CRF layer optimization. Therefore, we reduce the learning rate of the language model to 5e-5 and use a learning rate of 1e-4 for the CRF. The number of training epochs, early stopping, and validation performance are handled as before.

**Sense Classification**  For sense classification, the batch size is 16. We have two hidden layers of size 256 and 64 before the actual classification layer. Each layer is followed by a 0.3 dropout layer. The model is optimized by Adam and has a learning rate of 1e-4. The loss is a combination of coarse sense loss and fine sense loss in the ratio of 1 to 2. We also use early stopping after 3 iterations without performance gain. Performance is measured by the mean of coarse and fine sense macro F1.

## B  Frequent Sense Flow Patterns

In Table 9 and Table 8, we present a list of the 50 most frequent patterns, sense flows, and change flows, respectively. The Tables are sorted by the macro average relative frequency of all domains.

## C  Feature Importance

Linear regression models approximate linear relations between inputs and outputs such that $y_i =$ $x_0\theta_0 + \ldots + x_j\theta_j$ is computed for each class $i$ and each $x_j$ input pattern is multiplied by its corresponding weight $\theta_j$. We calculate the importance of each coefficient with respect to the learned class by $e^{\theta_i}$. Note that we omit the intercept in order to entirely focus on the linear combination of features. Figure 4 shows the 15 most important features per class.

| | Flow Pattern | Mean | AES | ACL | BBC | PEC | NYT | WSJ | MED | TED | UN |
|---|---|---|---|---|---|---|---|---|---|---|---|
| 1 | Concession-Conjunction | 4.26 | 2.88 | 3.75 | 5.45 | 6.03 | 3.71 | 6.15 | 3.63 | 3.27 | 3.48 |
| 2 | Conjunction-Concession | 4.03 | 3.00 | 2.02 | 5.28 | 5.17 | 3.81 | 6.48 | 3.57 | 3.41 | 3.55 |
| 3 | Conjunction-<pad> | 3.55 | 1.56 | 7.43 | 3.57 | 3.45 | 1.47 | 3.16 | 6.04 | 1.18 | 4.11 |
| 4 | Conjunction-Cause | 3.47 | 3.36 | 2.09 | 1.53 | 9.48 | 1.19 | 1.99 | 2.31 | 5.43 | 3.87 |
| 5 | Cause-Conjunction | 3.44 | 3.36 | 2.16 | 1.62 | 9.48 | 1.22 | 2.16 | 2.09 | 5.08 | 3.79 |
| 6 | <pad>-Conjunction | 2.75 | 0.96 | 4.04 | 2.64 | 1.72 | 1.24 | 2.82 | 5.99 | 1.18 | 4.19 |
| 7 | Conjunction-Concession-Conjunction | 1.78 | 0.96 | 0.94 | 2.47 | 1.72 | 1.80 | 2.66 | 1.59 | 1.74 | 2.13 |
| 8 | Condition-Conjunction | 1.75 | 4.56 | 0.58 | 1.28 | 3.45 | 0.91 | 1.50 | 0.66 | 1.67 | 1.18 |
| 9 | Synchronous-Conjunction | 1.60 | 0.96 | 1.37 | 1.79 | 1.72 | 1.70 | 1.99 | 1.04 | 1.81 | 2.05 |
| 10 | Conjunction-Condition | 1.58 | 3.96 | 0.65 | 1.28 | 2.59 | 0.90 | 1.33 | 0.71 | 1.53 | 1.26 |
| 11 | Conjunction-Asynchronous | 1.54 | 0.96 | 1.73 | 2.13 | 0.86 | 1.90 | 2.33 | 1.26 | 1.81 | 0.87 |
| 12 | Asynchronous-Conjunction | 1.50 | 0.72 | 1.66 | 2.47 | 0.86 | 1.90 | 2.16 | 1.21 | 1.67 | 0.87 |
| 13 | Conjunction-Cause-Conjunction | 1.46 | 1.20 | 0.94 | 0.60 | 3.45 | 0.47 | 0.66 | 0.93 | 2.72 | 2.21 |
| 14 | <pad>-Concession | 1.39 | 0.60 | 3.39 | 1.53 | 2.59 | 0.67 | 1.33 | 1.81 | 0.21 | 0.39 |
| 15 | Conjunction-Synchronous | 1.39 | 0.84 | 1.15 | 1.62 | 0.86 | 1.53 | 1.66 | 1.10 | 1.60 | 2.13 |
| 16 | Concession-Cause | 1.19 | 1.56 | 0.87 | 1.02 | 2.59 | 0.71 | 1.50 | 0.77 | 1.18 | 0.55 |
| 17 | Cause-<pad> | 1.09 | 0.84 | 0.58 | 0.60 | 2.59 | 0.38 | 0.83 | 1.32 | 1.60 | 1.03 |
| 18 | Cause-Concession | 1.05 | 1.32 | 0.50 | 0.94 | 2.59 | 0.65 | 1.33 | 0.44 | 1.18 | 0.47 |
| 19 | Concession-Conjunction-Concession | 0.98 | 0.60 | 0.36 | 1.79 | 0.86 | 1.24 | 1.99 | 0.60 | 0.77 | 0.63 |
| 20 | <pad>-Cause | 0.95 | 0.72 | 0.87 | 0.60 | 1.72 | 0.38 | 1.00 | 0.82 | 1.60 | 0.87 |
| 21 | Manner-Conjunction | 0.89 | 0.24 | 2.88 | 0.51 | 0.86 | 0.39 | 0.50 | 0.93 | 0.35 | 1.34 |
| 22 | Cause-Conjunction-Cause | 0.81 | 0.72 | 0.29 | 0.17 | 2.59 | 0.18 | 0.33 | 0.38 | 1.81 | 0.79 |
| 23 | Concession-Conjunction-<pad> | 0.81 | 0.24 | 1.37 | 1.19 | 0.86 | 0.45 | 1.00 | 1.48 | 0.14 | 0.55 |
| 24 | Concession-Condition | 0.78 | 2.04 | 0.22 | 0.85 | 0.86 | 0.59 | 1.16 | 0.38 | 0.56 | 0.32 |
| 25 | Concession-<pad> | 0.76 | 0.36 | 0.65 | 1.28 | 0.86 | 0.62 | 1.33 | 1.26 | 0.21 | 0.24 |
| 26 | Conjunction-Manner | 0.72 | 0.24 | 2.02 | 0.43 | 0.86 | 0.34 | 0.50 | 0.71 | 0.28 | 1.11 |
| 27 | <pad>-Concession-Conjunction | 0.65 | 0.24 | 1.73 | 0.77 | 0.86 | 0.33 | 0.50 | 1.10 | 0.07 | 0.24 |
| 28 | <pad>-Conjunction-Concession | 0.65 | 0.24 | 0.79 | 0.85 | 0.00 | 0.38 | 0.83 | 1.59 | 0.21 | 0.95 |
| 29 | Concession-Conjunction-Cause | 0.65 | 0.48 | 0.50 | 0.43 | 1.72 | 0.35 | 0.50 | 0.38 | 0.97 | 0.55 |
| 30 | Conjunction-Asynchronous-Conjunction | 0.65 | 0.36 | 0.87 | 0.85 | 0.00 | 0.89 | 0.83 | 0.60 | 0.97 | 0.47 |
| 31 | Conjunction-Condition-Conjunction | 0.63 | 1.56 | 0.22 | 0.51 | 0.86 | 0.35 | 0.50 | 0.22 | 0.70 | 0.71 |
| 32 | Purpose-Conjunction | 0.63 | 0.60 | 0.72 | 0.17 | 0.86 | 0.15 | 0.17 | 0.33 | 0.42 | 2.29 |
| 33 | Contrast-Conjunction | 0.61 | 0.36 | 0.43 | 0.51 | 0.86 | 0.26 | 0.66 | 1.92 | 0.21 | 0.24 |
| 34 | Conjunction-Contrast | 0.60 | 0.24 | 0.43 | 0.43 | 0.86 | 0.25 | 0.66 | 2.09 | 0.21 | 0.24 |
| 35 | Conjunction-Purpose | 0.60 | 0.48 | 0.58 | 0.26 | 0.86 | 0.15 | 0.17 | 0.16 | 0.35 | 2.37 |
| 36 | Cause-Conjunction-<pad> | 0.58 | 0.36 | 0.94 | 0.34 | 0.86 | 0.14 | 0.33 | 0.88 | 0.42 | 0.95 |
| 37 | Conjunction-Synchronous-Conjunction | 0.56 | 0.36 | 0.36 | 0.68 | 0.00 | 0.69 | 0.50 | 0.44 | 0.84 | 1.18 |
| 38 | Cause-Condition | 0.54 | 2.40 | 0.07 | 0.17 | 0.86 | 0.17 | 0.17 | 0.11 | 0.70 | 0.24 |
| 39 | Asynchronous-Concession | 0.53 | 0.24 | 0.14 | 1.36 | 0.00 | 0.82 | 1.33 | 0.33 | 0.35 | 0.16 |
| 40 | Condition-Concession | 0.52 | 1.56 | 0.14 | 0.68 | 0.00 | 0.51 | 1.00 | 0.16 | 0.49 | 0.16 |
| 41 | Condition-Cause | 0.51 | 2.28 | 0.07 | 0.17 | 0.86 | 0.15 | 0.17 | 0.11 | 0.56 | 0.24 |
| 42 | Concession-Synchronous | 0.50 | 0.36 | 0.29 | 0.85 | 0.00 | 0.81 | 1.16 | 0.33 | 0.49 | 0.24 |
| 43 | Cause-Conjunction-Concession | 0.50 | 0.48 | 0.14 | 0.34 | 0.86 | 0.36 | 0.66 | 0.27 | 0.97 | 0.39 |
| 44 | Concession-Asynchronous | 0.48 | 0.24 | 0.14 | 1.11 | 0.00 | 0.81 | 1.33 | 0.22 | 0.35 | 0.16 |
| 45 | Synchronous-Concession | 0.47 | 0.24 | 0.22 | 0.85 | 0.00 | 0.79 | 1.16 | 0.27 | 0.42 | 0.32 |
| 46 | Concession-Cause-Conjunction | 0.47 | 0.48 | 0.43 | 0.43 | 0.86 | 0.29 | 0.50 | 0.33 | 0.63 | 0.32 |
| 47 | Conjunction-Concession-<pad> | 0.46 | 0.12 | 0.36 | 0.77 | 0.86 | 0.34 | 0.66 | 0.71 | 0.14 | 0.16 |
| 48 | Instantiation-Conjunction | 0.41 | 0.12 | 0.14 | 0.00 | 2.59 | 0.11 | 0.33 | 0.00 | 0.21 | 0.16 |
| 49 | Conjunction-Cause-<pad> | 0.41 | 0.36 | 0.43 | 0.17 | 0.86 | 0.10 | 0.17 | 0.71 | 0.49 | 0.39 |
| 50 | Concession-Conjunction-Concession-Conjunction | 0.40 | 0.24 | 0.14 | 0.85 | 0.00 | 0.58 | 0.83 | 0.22 | 0.35 | 0.39 |

Table 8: Relative frequency of top 50 discourse *sense change flows* with n-grams of sizes 2 to 4. The table entries are sorted by their macro average over all corpora.

| | Flow Pattern | Mean | AES | ACL | BBC | PEC | NYT | WSJ | MED | TED | UN |
|---|---|---|---|---|---|---|---|---|---|---|---|
| 1 | Conjunction-Conjunction | 9.09 | 5.65 | 8.95 | 8.21 | 12.70 | 5.21 | 8.29 | 9.11 | 8.83 | 14.90 |
| 2 | Conjunction-Conjunction-Conjunction | 3.63 | 1.76 | 3.33 | 3.03 | 3.97 | 1.92 | 3.11 | 3.65 | 3.82 | 8.09 |
| 3 | Concession-Conjunction | 3.30 | 2.39 | 2.81 | 4.37 | 4.76 | 2.96 | 5.01 | 2.75 | 2.43 | 2.19 |
| 4 | Conjunction-Concession | 3.15 | 2.51 | 1.63 | 4.28 | 3.97 | 3.05 | 5.35 | 2.75 | 2.57 | 2.27 |
| 5 | Cause-Conjunction | 2.64 | 2.89 | 1.85 | 1.25 | 7.14 | 0.98 | 1.90 | 1.63 | 3.76 | 2.34 |
| 6 | Conjunction-Cause | 2.63 | 2.89 | 1.78 | 1.25 | 7.14 | 0.96 | 1.55 | 1.80 | 3.96 | 2.34 |
| 7 | Conjunction-<pad> | 2.37 | 1.26 | 5.18 | 2.50 | 2.38 | 1.03 | 2.25 | 3.82 | 0.83 | 2.12 |
| 8 | <pad>-Conjunction | 1.82 | 0.75 | 2.81 | 1.78 | 1.59 | 0.86 | 1.90 | 3.77 | 0.76 | 2.19 |
| 9 | Conjunction-Conjunction-Conjunction-Conjunction | 1.54 | 0.50 | 1.18 | 1.16 | 1.59 | 0.74 | 1.04 | 1.41 | 1.74 | 4.46 |
| 10 | Condition-Conjunction | 1.38 | 3.89 | 0.52 | 1.07 | 2.38 | 0.74 | 1.21 | 0.56 | 1.25 | 0.76 |
| 11 | Conjunction-Condition | 1.32 | 3.39 | 0.52 | 1.07 | 2.38 | 0.73 | 1.21 | 0.56 | 1.18 | 0.83 |
| 12 | Synchronous-Conjunction | 1.26 | 0.88 | 1.11 | 1.43 | 1.59 | 1.36 | 1.55 | 0.84 | 1.32 | 1.29 |
| 13 | Conjunction-Asynchronous | 1.23 | 0.88 | 1.48 | 1.69 | 0.79 | 1.50 | 1.90 | 1.01 | 1.32 | 0.53 |
| 14 | Conjunction-Concession-Conjunction | 1.21 | 0.75 | 0.67 | 1.61 | 1.59 | 1.18 | 1.73 | 1.07 | 1.11 | 1.21 |
| 15 | Asynchronous-Conjunction | 1.19 | 0.63 | 1.33 | 1.96 | 0.79 | 1.51 | 1.73 | 0.96 | 1.25 | 0.53 |
| 16 | Cause-Cause | 1.18 | 1.76 | 0.22 | 0.45 | 2.38 | 0.41 | 1.04 | 0.45 | 2.92 | 0.98 |
| 17 | Concession-Concession | 1.14 | 1.13 | 0.52 | 2.05 | 0.79 | 1.33 | 2.59 | 0.84 | 0.63 | 0.38 |
| 18 | Concession-Conjunction-Conjunction | 1.09 | 0.63 | 1.11 | 1.61 | 0.79 | 1.01 | 1.55 | 1.01 | 0.97 | 1.13 |
| 19 | Conjunction-Synchronous | 1.07 | 0.63 | 0.89 | 1.25 | 0.79 | 1.22 | 1.38 | 0.90 | 1.18 | 1.36 |
| 20 | Conjunction-Conjunction-<pad> | 1.06 | 0.38 | 2.74 | 1.07 | 0.79 | 0.41 | 0.86 | 1.80 | 0.28 | 1.21 |
| 21 | Conjunction-Conjunction-Concession | 1.02 | 0.63 | 0.37 | 1.43 | 0.79 | 1.04 | 1.73 | 1.01 | 1.04 | 1.13 |
| 22 | <pad>-Concession | 1.02 | 0.50 | 2.59 | 1.16 | 1.59 | 0.53 | 1.04 | 1.35 | 0.14 | 0.30 |
| 23 | Conjunction-Cause-Conjunction | 0.99 | 0.88 | 0.74 | 0.45 | 2.38 | 0.35 | 0.52 | 0.67 | 1.67 | 1.21 |
| 24 | Concession-Cause | 0.92 | 1.38 | 0.74 | 0.89 | 1.59 | 0.58 | 1.21 | 0.62 | 0.90 | 0.38 |
| 25 | Cause-Conjunction-Conjunction | 0.82 | 0.75 | 0.59 | 0.36 | 1.59 | 0.32 | 0.52 | 0.56 | 1.46 | 1.21 |
| 26 | Cause-Concession | 0.78 | 1.13 | 0.37 | 0.80 | 1.59 | 0.53 | 1.04 | 0.39 | 0.90 | 0.30 |
| 27 | Conjunction-Conjunction-Cause | 0.77 | 0.63 | 0.44 | 0.36 | 1.59 | 0.30 | 0.35 | 0.62 | 1.53 | 1.13 |
| 28 | <pad>-Conjunction-Conjunction | 0.69 | 0.13 | 0.89 | 0.80 | 0.00 | 0.33 | 0.69 | 1.74 | 0.28 | 1.36 |
| 29 | Manner-Conjunction | 0.69 | 0.25 | 2.37 | 0.36 | 0.79 | 0.32 | 0.35 | 0.73 | 0.21 | 0.83 |
| 30 | Concession-Condition | 0.64 | 1.76 | 0.22 | 0.71 | 0.79 | 0.48 | 0.86 | 0.34 | 0.42 | 0.15 |
| 31 | Concession-<pad> | 0.61 | 0.38 | 0.52 | 0.98 | 0.79 | 0.49 | 1.04 | 0.96 | 0.14 | 0.15 |
| 32 | Conjunction-Manner | 0.58 | 0.25 | 1.70 | 0.36 | 0.79 | 0.28 | 0.35 | 0.56 | 0.21 | 0.68 |
| 33 | Cause-<pad> | 0.58 | 0.63 | 0.44 | 0.36 | 0.79 | 0.22 | 0.52 | 0.96 | 0.83 | 0.45 |
| 34 | <pad>-Cause | 0.55 | 0.63 | 0.67 | 0.36 | 0.79 | 0.23 | 0.52 | 0.56 | 0.83 | 0.38 |
| 35 | Concession-Conjunction-Concession | 0.53 | 0.38 | 0.15 | 0.89 | 0.79 | 0.64 | 1.04 | 0.34 | 0.28 | 0.23 |
| 36 | Contrast-Conjunction | 0.51 | 0.25 | 0.37 | 0.45 | 0.79 | 0.21 | 0.69 | 1.52 | 0.14 | 0.15 |
| 37 | Purpose-Conjunction | 0.48 | 0.50 | 0.59 | 0.18 | 0.79 | 0.12 | 0.17 | 0.22 | 0.28 | 1.44 |
| 38 | Conjunction-Contrast | 0.46 | 0.13 | 0.37 | 0.27 | 0.79 | 0.20 | 0.52 | 1.57 | 0.14 | 0.15 |
| 39 | Conjunction-Conjunction-Conjunction-<pad> | 0.46 | 0.13 | 1.33 | 0.45 | 0.00 | 0.18 | 0.35 | 0.84 | 0.14 | 0.76 |
| 40 | Cause-Condition | 0.46 | 2.01 | 0.07 | 0.18 | 0.79 | 0.14 | 0.17 | 0.11 | 0.49 | 0.15 |
| 41 | Condition-Concession | 0.45 | 1.38 | 0.15 | 0.54 | 0.00 | 0.42 | 0.86 | 0.17 | 0.35 | 0.15 |
| 42 | Conjunction-Condition-Conjunction | 0.45 | 1.13 | 0.15 | 0.36 | 0.79 | 0.26 | 0.35 | 0.17 | 0.49 | 0.38 |
| 43 | Conjunction-Purpose | 0.44 | 0.50 | 0.44 | 0.18 | 0.79 | 0.12 | 0.17 | 0.11 | 0.21 | 1.44 |
| 44 | Condition-Cause | 0.44 | 2.01 | 0.07 | 0.09 | 0.79 | 0.13 | 0.17 | 0.11 | 0.42 | 0.15 |
| 45 | Conjunction-Asynchronous-Conjunction | 0.43 | 0.25 | 0.67 | 0.54 | 0.00 | 0.61 | 0.52 | 0.39 | 0.63 | 0.30 |
| 46 | Concession-Asynchronous | 0.43 | 0.25 | 0.15 | 0.98 | 0.00 | 0.66 | 1.21 | 0.22 | 0.28 | 0.08 |
| 47 | Concession-Synchronous | 0.42 | 0.25 | 0.30 | 0.71 | 0.00 | 0.67 | 1.04 | 0.28 | 0.35 | 0.15 |
| 48 | Asynchronous-Concession | 0.41 | 0.13 | 0.15 | 1.16 | 0.00 | 0.67 | 1.04 | 0.28 | 0.21 | 0.08 |
| 49 | Synchronous-Concession | 0.41 | 0.25 | 0.22 | 0.71 | 0.00 | 0.65 | 1.04 | 0.22 | 0.35 | 0.23 |
| 50 | Condition-Conjunction-Conjunction | 0.40 | 1.01 | 0.15 | 0.36 | 0.79 | 0.24 | 0.17 | 0.11 | 0.42 | 0.38 |

Table 9: Relative frequency of top 50 discourse *sense flows* with n-grams of sizes 2 to 4. The table entries are sorted by their macro average over all corpora.

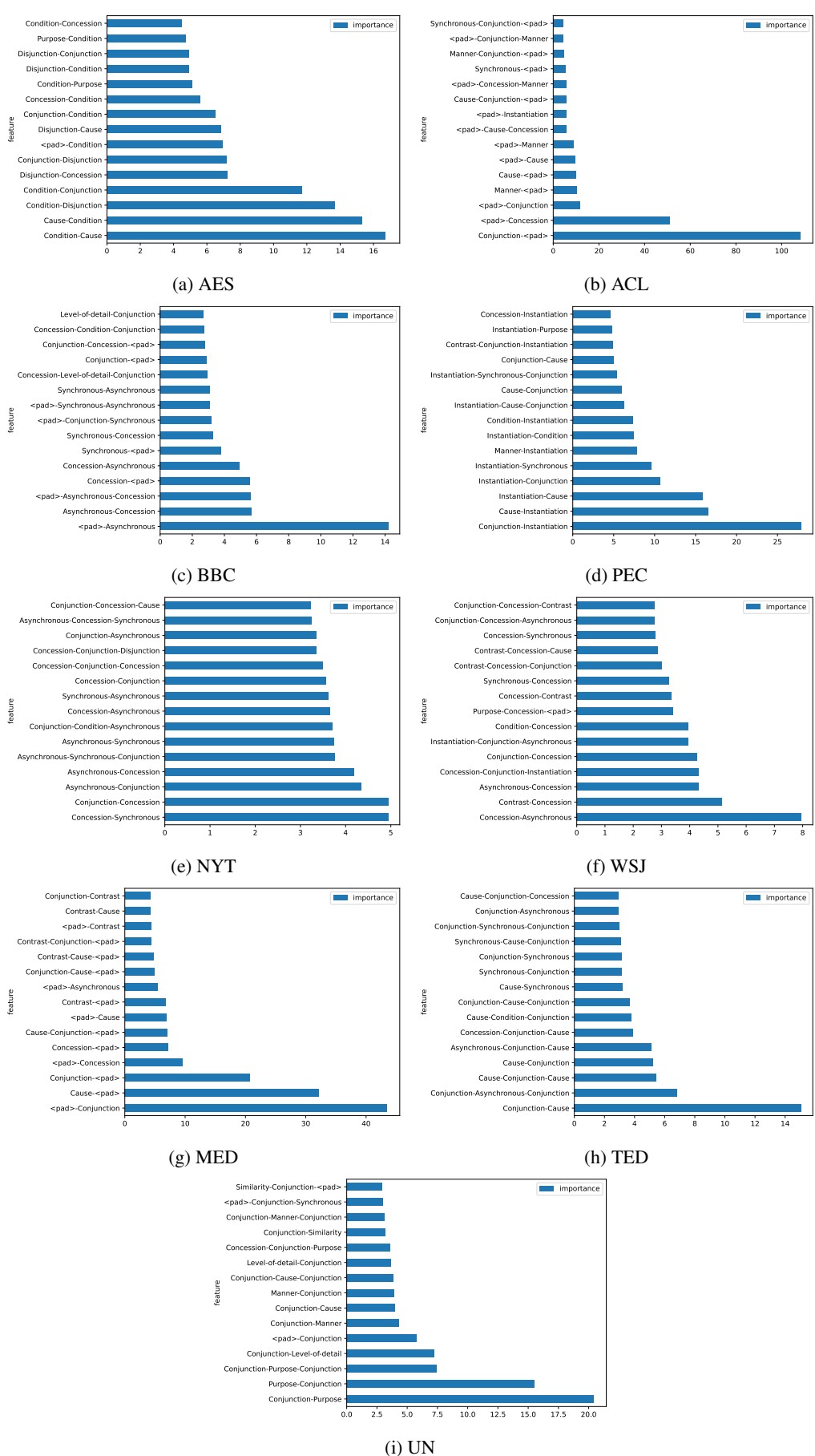

Figure 4: Feature importance plots for genre classification.

# D Differences of Connectives and their Modifications

In the following, we automatically compile a table of connectives from the explicit relations we identified in the corpora. Similar to the annotation manual (Webber et al., 2019), we show the connective heads (left) and their corresponding modified occurrences (right) along with the counts. Here, we exclude all modified connectives identified in the original PDTB3 data. Due to the automatic mapping from modified occurrences to their possible heads, errors involve mismatches and non-explicit signals that have been identified.

| | | |
|---|---|---|
| about | 2 | about time (1), about the same that (1) |
| after | 712 | not long after (65), the day after (48), two years after (46), hours after (44), days after (39), four years after (29), just days after (29), three days after (28), moments after (27), six months after (27), just hours after (25), four months after (22), five months after (18), three weeks after (16), one week after (16), a few years after (15), five days after (15), weeks after (14), one year after (14), well after (13), one month after (13), eight years after (13), just weeks after (12), six days after (12), seconds after (11), a decade after (11), several months after (11), nine years after (11), just months after (10), only days after (9), less than a year after (9), an hour after (9), less than a week after (9), not after (8), 20 years after (8), six weeks after (8), in the days after (8), less than 24 hours after (8), only hours after (7), just a week after (7), ten days after (7), two hours after (7), decades after (7), a night after (7), sometime after (6), less than two years after (6), just two years after (6), about a month after (6), just four days after (6), just a few months after (6), [. . .] |
| afterward | 10 | immediately afterward (12), only afterward (3), not long afterward (3), long afterward (2), even afterward (1), right afterward (1), a week afterward (1), half an hour afterward (1), before and afterward (1), very soon afterward (1) |
| afterwards | 4 | immediately afterwards (3), soon afterwards (1), not long afterwards (1), right afterwards (1) |
| also | 4 | not also (5), even also (1), only also (1), so also (1) |
| and | 4 | alternately and (7), simultaneously and (2), either and (1), at one and the same time (1) |
| as | 84 | as a consequence (69), as as (66), as such (25), as opposed to (24), only as (19), as fact (16), as an example (12), inasmuch as (11), as example (8), as fast as (7), as to (5), as a bonus (5), right as (5), exactly as (4), as soon (4), almost as (4), as far as (3), at least as (3), soon as (3), as quickly as (3), not as (3), often as (3), precisely as (3), as long (3), immediately as (2), as of fact (2), as result (2), as early as (2), perhaps as (2), somewhat as (2), first as (2), as bonus (2), just as as (2), as an instance (1), where as (1), at long as (1), as long a (1), while the same time as (1), as all as (1), just as fast as (1), both as (1), almost precisely as (1), for as (1), a as (1), anxiety as (1), because as (1), as long that (1), almost as quickly as (1), as a well as (1), hard as (1), [. . .] |
| as a result | 4 | possibly as a result (2), partly as a result (2), perhaps as a result (1), presumably as a result (1) |
| as if | 3 | almost as if (8), just as if (4), much as if (1) |
| as long as | 5 | just as long as (6), as long as then (1), at least as long as (1), as long as three months after (1), for as long as (1) |
| as soon as | 3 | almost as soon as (23), nearly as soon as (1), perhaps as soon as (1) |
| as though | 1 | almost as though (1) |
| as well as | 1 | both as well as (2) |
| at the same time | 3 | at the same time as (5), all at the same time (2), almost at the same time (2) |
| because | 66 | precisely because (57), mostly because (46), not least because (44), probably because (37), maybe because (32), not just because (23), possibly because (19), either because (16), all because (16), solely because (11), both because (10), specifically because (8), in small part because (7), not so much because (7), often because (7), whether because (6), in large measure because (6), part because (6), at least in part because (5), not necessarily because (5), chiefly because (4), usually because (4), sometimes because (4), most likely because (3), more because (3), purely because (3), almost because (2), supposedly because (2), not simply because (2), ostensibly because (2), reportedly because (2), less because (2), some because (2), for because (1), because than (1), no because (1), first off because (1), initially because (1), most notably because (1), not merely because (1), obviously because (1), necessarily because (1), generally because (1), many because (1), perhaps in part because (1), principally because (1), almost certainly because (1), adventurous because (1), at in part because (1), in no small part because (1), [. . .] |
| because of | 3 | all because of (2), supposedly because of (1), in part because of (1) |

| before | 303 | not before (53), well before (48), hours before (33), a year before (25), right before (24), moments before (19), days before (18), months before (17), a few days before (17), just hours before (16), minutes before (15), not long before (14), just weeks before (13), a month before (12), a day before (12), four days before (12), two weeks before (12), weeks before (12), one day before (11), the day before (11), a few months before (11), an hour before (10), three days before (10), six months before (10), two hours before (9), three months before (9), a few hours before (8), only days before (7), the year before (7), three years before (6), in the months before (6), soon before (6), four months before (6), just minutes before (5), 10 days before (5), five years before (5), five months before (5), three weeks before (5), five days before (5), way before (5), 30 minutes before (4), less than two months before (4), six days before (4), several days before (4), a full year before (4), one year before (4), four years before (4), about a month before (4), less than two weeks before (4), less than a week before (4), [. . . ] |
|---|---|---|
| before and after | 3 | just before and after (2), both before and after (1), immediately before and after (1) |
| both and | 2 | both because and (1), both and also (1) |
| but | 11 | not but (29), only but (4), not merely but (4), just but (3), not so much but (2), not along but (1), not simply but (1), merely but (1), perhaps but (1), instead but (1), but example (1) |
| but also | 6 | not but also (10), only but also (3), not simply but also (3), not merely but also (3), just but also (1), both but also (1) |
| by | 59 | by the same token (31), either by (26), in part by (23), perhaps by (18), mainly by (14), primarily by (13), not by (13), possibly by (10), especially by (9), first by (8), often by (8), usually by (7), whether by (5), specifically by (4), presumably by (3), both by (3), in large part by (3), by extension (2), notably by (2), most by (2), part by (2), in by (2), generally by (2), most notably by (2), precisely by (2), say by (2), by end (2), less by (2), apparently by (2), if by (2), principally by (2), by the end (2), by analogy (2), including by (2), essentially by (1), even by (1), sometimes by (1), by example (1), basically by (1), by gum (1), by contract (1), by the time (1), now by (1), typically by (1), by sharp contrast (1), by time (1), lately by (1), visibly by (1), by extreme contrast (1), definitively by (1), [. . . ] |
| earlier | 2 | a moment earlier (1), two earlier (1) |
| either or | 2 | either or else (3), either because or (1) |
| else | 2 | or else (47), either else (1) |
| even after | 1 | not even after (3) |
| even before | 2 | perhaps even before (1), sometimes even before (1) |
| even if | 1 | not even if (1) |
| even when | 1 | not even when (8) |
| for | 9 | for comparison (8), for starters (4), just for (4), for an example (3), simply for (2), partly for (2), for instant (1), for tho' they (1), for contrast (1) |
| from | 5 | apart from (2), not from (2), even from (1), from the same time (1), 10 months from (1) |
| given | 2 | especially given (6), particularly given (2) |
| if | 36 | if than (59), not if (29), if not (12), at least if (7), regardless if (5), if and only if (3), either if (2), exspecially if (2), and if (2), perhaps if (2), maybe if (2), simply if (2), even it if (2), just if (2), certainly if (2), at least not if (2), irrespective if (1), for if (1), despite if (1), instead if (1), whether if (1), rather if (1), or rather if (1), than if (1), crucial if (1), almost if (1), like if (1), primarily if (1), or if (1), now if (1), particuary if (1), badly if (1), old if (1), / if (1), if anything (1), sorry if (1) |
| if only | 4 | if only because (43), if only by (3), if only so (2), now if only (1) |
| if then | 1 | especially if then (1) |
| in | 44 | in conclusion (738), in summary (32), in essence (11), in parallel (10), in between (8), in the other hand (7), in brief (6), in the same way (6), in total (5), in consequence (4), in retrospect (4), as in (4), in the interim (4), in stark contrast (4), in effect (4), in opposite (3), in example (3), in result (3), in word (3), even in (3), in the alternative (3), in the same breath (3), in reality (2), in other word (2), in a word (2), in the same time (2), in spite of (2), in sharp contrast (2), in words (2), in away (1), in specific (1), in vivid contrast (1), in compensation (1), in the contrary (1), in which case (1), in of (1), in future (1), only in (1), in truth (1), in the mean time (1), in sense (1), in closing (1), in the same manner (1), in the same token (1) |
| in addition | 1 | in addition to (2) |
| in case | 1 | especially in case (1) |
| in order | 2 | just in order (2), in order for (1) |

| | | |
|---|---|---|
| in particular | 5 | in particular when (7), in particular by (4), in particular because (3), in particular as (3), in particular since (2) |
| in the meantime | 1 | at in the meantime (1) |
| instead | 1 | if instead (2) |
| later | 72 | a later (48), a short time later (14), two later (12), three later (11), a year later (10), moments later (10), only later (10), a minute later (9), hours later (8), an hour later (7), two years later (5), two minutes later (5), a month later (5), minutes later (5), four years later (4), several later (4), five later (3), six months later (3), an later (3), two hours later (3), two months later (3), a years later (3), a time later (2), a few minutes later (2), a few days later (2), nine later (2), a minutes later (2), four months later (2), three months later (2), several hours later (2), five minutes later (2), about an hour later (2), four later (2), a moment later (2), three years later (2), a months later (2), a few hours later (2), a short later (1), less than half an hour later (1), less than a minute later (1), minute later (1), much later (1), 50 minutes later (1), a few weeks later (1), seven later (1), a few months later (1), years later (1), several minutes later (1), a bit later (1), a day later (1), [. . .] |
| like | 10 | just like (137), much like (19), almost like (8), kind of like (2), kind like (2), a bit like (2), rather like (1), somewhat like (1), pretty much like (1), or like (1) |
| meantime | 1 | in meantime (1) |
| not just but | 1 | not just because but (1) |
| not only | 13 | not only because (65), not only and (7), not only when (4), not only by (4), not only while (2), not only if (2), not only or (1), not only it (1), not only then (1), not only that (1), not only indeed (1), not only since (1), not only as (1) |
| not only but | 1 | not only because but (1) |
| now that | 3 | especially now that (10), even now that (3), particularly now that (1) |
| on | 27 | on the other (11), on the flip side (8), on the other side (7), on one hand on the other (5), further on (3), on the other end (3), on the same note (2), on other hand (2), on the other hands (2), based on (2), on the another hand (1), on contrary (1), on the other had (1), on one hand the other (1), on the other extreme (1), on one on the other side (1), on the upside (1), late on (1), on account (1), on one hand (1), immediately on (1), on the downside (1), on the completely other side (1), on the opposite (1), on on the other (1), on side (1), earlier on (1) |
| on the one hand | 2 | on the one hand (16), on the one hand the other (4) |
| on the other hand | 1 | on one hand on the other hand (6) |
| once | 8 | even once (7), only once (5), especially once (4), at once (3), particularly once (2), once again (1), at least once (1), all at once (1) |
| only | 5 | only that (2), only through (2), only cause (1), only order (1), only ifit (1) |
| or | 6 | whether or not (27), / or (22), whether or (2), and or (1), eithers or (1), at the same or (1) |
| otherwise | 1 | so otherwise (1) |
| rather | 1 | rather elders (1) |
| rather than | 2 | rather than by (1), rather than because (1) |
| simultaneously | 1 | and simultaneously (1) |
| since | 33 | not since (16), at least since (9), almost since (5), years since (5), probably since (3), two years since (2), months since (2), decades since (1), a little over a year since (1), even since (1), a century since (1), eight years since (1), before and since (1), only since (1), half since (1), all since (1), five years since (1), two months since (1), roughly since (1), hours since (1), both since (1), a half since (1), almost a year since (1), in since (1), ill since (1), more than two years since (1), in two weeks since (1), two days since (1), a half years since (1), in months since (1), almost two months since (1), ten years since (1), six years since (1) |
| so | 15 | just so (48), in part so (3), so way (2), not so (2), mostly so (2), only so (2), all so (2), so the (1), precisely so (1), so long (1), specifically so (1), mainly so (1), largely so (1), not just so (1), especially so (1) |
| so as | 1 | not so as (1) |
| so that | 5 | so that way (5), just so that (5), largely so that (1), presumably so that (1), often so that (1) |
| specifically | 1 | more specifically (56) |
| still | 1 | later still (1) |
| that is | 3 | that is say (3), that is to say (2), that is if (1) |

| then | 25 | then again (112), only then (72), unless then (14), once then (10), just then (10), rather then (7), since then (6), right then (6), so then (4), until then (4), becuase then (3), because then (3), before then (2), in then end (1), i then (1), due then (1), it then (1), from then (1), an then (1), any then (1), anytime then (1), for then (1), other then (1), although then (1), at then (1) |
|---|---|---|
| thereafter | 2 | soon thereafter (9), as soon thereafter (1) |
| though | 3 | event though (1), what though (1), similar though (1) |
| thus | 1 | only thus (1) |
| till | 2 | at least till (2), not till (1) |
| unless | 2 | not unless (7), until and unless (1) |
| until | 9 | not until (22), unless and until (8), or until (2), up until (2), perhaps until (2), right up until (2), often until (1), unless or until (1), supposedly until (1) |
| upon | 3 | immediately upon (2), almost upon (1), only upon (1) |
| when | 62 | not when (34), right when (13), perhaps when (5), mostly when (5), both when (4), immediately when (4), starting when (3), specially when (3), mainly when (3), except when (3), where and when (3), as and when (3), preferably when (2), expecially when (2), in when (2), apparently when (2), not just when (2), at when (2), notably when (2), simply when (2), either when (2), sometimes when (2), now when (2), precisely when (2), exactly when (2), also when (1), specifically when (1), even sometimes when (1), exspecially when (1), primarily when (1), less than 24 hours when (1), especially not when (1), most famously when (1), all when (1), especially now when (1), most commonly when (1), beginning when (1), around when (1), more when (1), a week when (1), ideal when (1), generally when (1), built when (1), most when (1), rarely when (1), often when (1), necessarily when (1), most recently when (1), initially when (1), certainly when (1), [. . . ] |
| when and if | 1 | only when and if (1) |
| whenever | 1 | wherever and whenever (1) |
| where | 5 | particularly where (7), especially where (4), even where (2), only where (1), when and where (1) |
| whether | 1 | whether not (1) |
| while | 14 | all the while (53), all while (28), only while (4), often while (3), both while (2), especially while (2), at least while (2), not while (2), particularly while (1), apparently while (1), preferably while (1), sometimes while (1), all this while (1), just while (1) |
| with | 14 | what with (12), not with (9), only with (6), albeit with (2), sometimes with (2), as with (1), with that (1), often with (1), because with (1), at not with (1), at least with (1), literally with (1), perhaps with (1), especially now with (1) |
| without | 15 | even without (27), not without (6), all without (6), with without (2), apparently without (2), preferably without (2), usually without (1), only without (1), especially without (1), reportedly without (1), always without (1), again without (1), also without (1), simply without (1), almost without (1) |

# E   Alternative Lexicalizations and their associated Senses

Below we present a selected number of alternative lexicalizations grouped by their predicted sense. We give exact occurrences in parentheses, while the second column presents the number of variants.

| | | |
|---|---|---|
| Comparison.Concession | 35 | the only difference is that (5), it's as simple as that (2), but my point is that (1), it's as easy as that (1), what I am trying to say (1), with all that has been said (1), the biggest difference is that (1), the main difference is that (1), what is unusual is that (1), what has changed is that (1), what is different this time is that (1), with the only difference being that (1), that of course was when (1), the one difference is that (1), one significant difference is that (1), the only difference being that (1), the only change the years is that (1), the difference that game was (1), the difference now is that (1), what is different is that (1), that's kind of the point (1), but the point is that (1), all of this to say (1), all of which is to say (1), it's a combination of this (1), what this is actually saying is that (1), what's interesting is that (1), I think the point is that (1), what's happening here is (1), what was so funny about it was that (1), the only difficulty is that (1), the thing about that was (1), what's different now is (1), that shift in perspective is like (1), what is different today is that (1) |
| Comparison.Contrast | 41 | this is in contrast to (5), that is significantly lower than (2), this was a significant jump (1), this is in stark contrast with (1), compared that new cost with (1), that's 30 percent higher than (1), that number is much higher than (1), the number is down from (1), the same is true for (1), likened the voting haiti to (1), that figure is comparable to (1), the figure was up from (1), at the other end of the spectrum (1), that is far higher than (1), the quarterly growth rate compared with (1), this "conversion rate" compares with (1), the inflation rate compares to (1), the difference this time was that (1), that is very different from (1), that is almost as many as (1), this is a change from (1), that is a dramatic shift (1), that figure is slightly higher than (1), that total was significantly down from (1), that is in contrast to (1), the only difference is that (1), that is a major increase from (1), that number is roughly the same as (1), that is in comparison to (1), a major difference is that (1), these results are similar to (1), this is in strong contrast to (1), this behavior is in contrast to (1), this is in sharp contrast to (1), which is in stark contrast to (1), that's as opposed to (1), what you 'll of all (1), it's a little bit like (1), the research squared up with (1), what makes this claim different is (1), this is in line with (1) |
| Comparison.Similarity | 8 | the same is true when (1), it's the same as (1), the whole effect is a little like (1), the same thing goes for (1), it's a bit like (1), this is kind of like (1), that's the same as (1), this is very similar to (1) |
| Contingency.Cause | 1268 | the purpose of this study was (20), the reason for that is (15), the reason for this is (12), part of the reason is (7), this is partly due to (6), this is the reason why (5), the purpose of this study is (5), what this means is that (5), as a result of that (5), this is largely due to (4), that is the reason why (4), this may be due to (4), this is mainly due to (4), this is one reason why (3), that is one reason why (3), what I mean by that is (3), one reason for this is (3), what that means is that (3), that was the reason for (3), it's enough to make (3), the result of that is (3), one of the reasons is (3), the reason I say this is because (2), that's one reason why (2), the goal of this task is (2), the key innovation is that (2), the goal of this paper is (2), this has given rise to (2), this situation has led to (2), the rise is due part to (2), popularity was behind the growth (2), the popularity of the net has meant that (2), the growth has been fuelled by (2), part of the impetus comes from (2), crucial to this has been (2), putting the drop down to (2), was perfect example of this (2), these rumours were fuelled by (2), this is likely to mean (2), this is one of the reasons why (2), that is in part because (2), that is one reason that (2), the delay was caused by (2), this is due in part to (2), some of that can be attributed to (2), that set the stage for (2), one way to do this is (2), the reason for it is (2), the no. 1 reason is (2), which may help explain why (2), [. . . ] |
| Contingency.Condition | 2 | in order to do that (2), the more and more and more (1) |
| Contingency.Purpose | 11 | that is one simple way (1), this is done in order (1), the plan is intended to (1), the decision appears to be an effort (1), as if to prove his point (1), the statement appeared to be an effort (1), in what was an effort (1), the purpose of this study was (1), the purpose of this was (1), the aim of the study is (1), another way to look at this is (1) |

| | | |
|---|---|---|
| Expansion.Conjunction | 143 | that is in addition to (7), this is in addition to (4), it all adds up to (3), this is not to mention (2), what is worse is that (2), what I'm saying is (1), which leads me to my next point (1), that brings up the subject (1), a salient feature is that (1), addition to the cs phenomenon (1), this aid comes on top of (1), the other key ingredient was (1), the most important of all is (1), another important point is that (1), one of the difficulties was (1), not to mention the fact (1), is in the same camp (1), what this is about is that (1), putting aside the obvious point that (1), compounding the complexity is that (1), that's not to mention (1), are doing the same thing (1), the same downward trend applies to (1), what is more surprising is that (1), what is stranger is that (1), is a case in point (1), a common thread in these efforts is (1), leave aside the fact that (1), the same is true of (1), what's worse is that (1), included in that run was (1), in addition to the forest sale (1), the increase comes on top of (1), what made matters worse was that (1), but real bottom line was (1), more to the contemporary point (1), that is on top of (1), what was worse was that (1), the high point of that trip was (1), what is known is that (1), a third benefit is that (1), the only real difference is that (1), that settlement was on top of (1), the only problem was that (1), it is as simple as that (1), what is true is that (1), all that could be said (1), that comes on top of (1), another factor in the fight (1), the issue of course is (1), [...] |
| Expansion.Instantiation | 80 | is a case in point (11), a case in point is (4), one example of this is (3), one of these things is (2), to take just one example (2), as an example of this (2), the most recent example of this was (2), it is as simple as that (1), that is just one example how (1), another bad example would be (1), this is one example of (1), a good example of this is (1), is another example of this (1), one of those freedoms is (1), a prime example of this is (1), an example for this is (1), an example of this would be (1), a paradigmatic example of this situation is (1), another trend this year has been (1), another example of this is (1), as a case in point (1), an obvious example of this happens (1), one such deal under consideration (1), among those he cited was (1), a case in point was (1), are just a few examples (1), the most notable example has been (1), one result of which was (1), one example of that is (1), the main one is that (1), an example of the first is (1), the best known example is (1), a recent case in point involves (1), the latest example of this is (1), a wonderful example here is (1), one of the best examples is (1), the most publicized example was (1), exhibit a has to be (1), at the top of that list (1), is the most literal example (1), one of the biggest is (1), among the strategies employed are (1), the single most salient fact is (1), be a case in point (1), the most frequent criticism has been (1), is only the most recent example (1), another way to look at it is (1), the setup is as follows (1), the most common one is (1), the most notorious examples were (1), [...] |
| Expansion.Level-of-detail | 42 | what I am trying to say is (1), this is especially relevant for (1), a clear cut example that supports this idea is (1), is a case in point (1), what is implied is that (1), one of many cases in point involves (1), it was as simple as that (1), I my point is that (1), the main point is that (1), the big story is that (1), the list so far includes (1), what's happening is that (1), the whole idea here is (1), the larger view here is (1), what is unusual is that (1), what is new is that (1), what is clear is that (1), is the most dramatic case in point (1), the one noticeable difference was that (1), what's new is that (1), what it comes down to is (1), this is particularly relevant for (1), a major theme to emerge (1), what I that is that (1), the main point here is (1), but the point here is (1), the whole point is that (1), what I mean by that is (1), but the point is that (1), and what it was is (1), what's fantastic is that (1), what this is is that (1), what is striking about this is that (1), the general point is that (1), but the basic thing is (1), what was amazing is that (1), what I mean by this is (1), what she was telling me is that (1), examples of decisions taken include (1), the most vivid example of that is the fact (1), what is important is that (1), is the most tangible proof of that (1) |
| Temporal.Asynchronous | 30 | at the end of the day (4), in the years since that day (2), this is usually followed by (2), this was the beginning of (2), at the end of that (2), after you get done with that (1), the news comes a week after (1), this was followed the same by (1), this was to be followed by (1), that meeting came a day after (1), the deal comes days after (1), that came a week after (1), after that process was completed (1), it was the beginning of (1), their removal came a week after (1), it was at that point (1), their announcement came one day after (1), that was the start of (1), its statements came hours after (1), her comments came a week after (1), his comments came a day after (1), the expansion into asia follows (1), the move comes just weeks after (1), at the end of that process (1), once I put that in context (1), now that that's done (1), that's usually followed by (1), what happened next was that (1), that was the point that (1), after that has gone through (1) |
| Temporal.Synchronous | 13 | as he has done so (1), as I was doing so (1), this comes in the face (1), while I'm doing that (1), in nearly the same period (1), at the time of the sale (1), while you're about it (1), while this was all going on (1), in the course of this (1), the next step in the process that (1), in the process of this occupation (1), the the point is that (1), an essential step in that regard (1) |