# OpenReview forum: "Discourse Sense Flows: Modelling the Rhetorical Style of Documents across Various Domains"
_EMNLP/2023/Conference — EMNLP 2023 Findings_

### Official Review · Reviewer_zypT · 2023-07-27

**Soundness:** 4

**Excitement:**

4: Strong: This paper deepens the understanding of some phenomenon or lowers the barriers to an existing research direction.

**Paper Topic And Main Contributions:**

This paper develops new models to extract explicit and alternatively lexicalized signals of discourse structure as well as their senses. These models are subsequently applied to raw corpora of various genres and domains. The notion of sense flows is introduced and compared across various genres/domains. One of the main findings is that n-gram patterns are stronger predictors than unigram distributions.

**Questions For The Authors:**

The paper seems to provide examples of one AltLex type; did the authors also deal with types that are syntactically and lexically flexible, e.g. "that compares with, the increase was due mainly to, once triggered", etc. Whichever type was targetted, it would be nice to see a sample of the extracted forms (particularly AltLexes) in an appendix.

**Reasons To Accept:**

This is a nicely presented paper with several merits. Firstly, the authors use the PDTB labels to identify not only explicit signals but also alternative lexicalizations, which are known to have free forms. Secondly, the concept of sense flows (basically discourse senses that linearly follow one another) is introduced. Although this notion is not entirely novel, using them to identify rhetorical patterns of different genres/domains is quite new. Another novelty of the paper is that the authors use both the PDTB sense hierarchy and a simplified hierarchy, excarding less frequent labels involving belief and speech act. The models are applied to both hierarchies as well as their combination and compared. As most work on discourse, the authors target the first and second level of the PDTB sense hierachy and train both sets of senses jointly.

The background is well-described, and the pipeline is clearly explained, enabling the replication of this work.

The results are also good, especially regarding alternative lexicalizations, which are obviously more difficult.

**Reasons To Reject:**

I don't have any reasons to reject this paper.

**Reproducibility:**

4: Could mostly reproduce the results, but there may be some variation because of sample variance or minor variations in their interpretation of the protocol or method.

**Reviewer Confidence:**

4: Quite sure. I tried to check the important points carefully. It's unlikely, though conceivable, that I missed something that should affect my ratings.

---

> ### Author Rebuttal · Authors · 2023-08-28
>
> We thank the reviewer for the feedback and agree this is definitely an interesting question.
>
> We neglected the list of AltLexes in the appendix due to their high variance, but we could easily provide some insides, e.g. grouped by its classified sense similar to the PDTB3 manual with an additional corpus source. However, we would prefer to focus on a subset, in particular, larger AltLexes to limit the used space. Also, these longer signals should be more interesting as they are usually more flexible and hard to catch with rule-based systems.

---

### Official Review · Reviewer_jZkU · 2023-08-04

**Soundness:** 3

**Excitement:**

3: Ambivalent: It has merits (e.g., it reports state-of-the-art results, the idea is nice), but there are key weaknesses (e.g., it describes incremental work), and it can significantly benefit from another round of revision. However, I won't object to accepting it if my co-reviewers champion it.

**Missing References:**

There has been earlier work on creating and using N-gram models of relations. The authors are refered to:

Or Biran and Kathleen McKeown. 2015. Discourse Planning with an N-gram Model of Relations. In Proceedings of the 2015 Conference on Empirical Methods in Natural Language Processing, pages 1973–1977, Lisbon, Portugal. Association for Computational Linguistics.

The authors should also refer to the The Penn Discourse Treebank 3.0 Annotation Manual, which documents connectives and their modifiers in Appendix B ("Explicit Connectives and their Modifiers"), rather than implying that they are the first to carry out such an analysis. They imply the same in Section 4.2, talking about modifiers of explicit connectives.  Both of these should be corrected.
The authors can get access to the annotation manual through documentation at the LDC. One does not need to have purchased the corpus to access the documentation at https://catalog.ldc.upenn.edu/docs/LDC2019T05/

**Paper Topic And Main Contributions:**

The authors' claimed contributions are (i) developing novel models for classifying senses of explicit discourse connectives and AltLex phrases; (ii) characterizing patterns of discourse sense N-grams (for low N) across different genres and domains; and (iii) determining whether such sense n-grams allow them to distinguish the rhetorical style of a document.

**Questions For The Authors:**

A. The authors claim (lines 219-220) that they "use our own predicted 219 discourse signals rather than the gold annotations [in the PDTB-3, which has the largest human-annotated set of annotated explicit and altlex relations]. The authors should tell us what labels from their set are assigned to the PDTB-3 annotations, and where this does and doesn't make sense.

B. Shi and Demberg (2017) have shown the extent of variability in the Penn TreeBank and how this argues for using cross validation (and the error brackets this allows), rather than a held-out test set. Can the authors comment on how their use of the latter (rather than cross-validation) might call the own results into question?

@inproceedings{shi-demberg-2017-need,
  title = "On the Need of Cross Validation for Discourse Relation Classification",
  author = "Shi, Wei  and Demberg, Vera",
  booktitle = "Proceedings of the 15th Conference of the {E}uropean
  Chapter of the Association for Computational Linguistics: Short Papers",
     month = apr,
     year = "2017",
     address = "Valencia, Spain",
     publisher = "Association for Computational Linguistics",
     pages = "150--156",
}

C. The authors claim (lines 367-368) that "Comparing both sense hierarchies, we notice an increased accuracy in our simplified hierarchy compared to the original one." This is disingenuous, given that (i) in terms of "coarse" senses, the model trained on the original sense labels does either better or equal to a model trained on the authors' simplified senses, and (ii) for the "fine" senses, they fail to indicate when any difference in performance is signficant.  Can they explain why this claim should not be corrected?

D. The authors' Appendix D ("Differences of Connectives and their Modifications") implies that "about", "in", "on", "only" etc. are connectives when they are merely part of the modifiers of other connectives. For example, contrary to what the authors claim, "in" is not a connective in the phrase "in conclusion" or "in summary" or "in contrast". In these constructions, the whole prepositional phrase is a connective, not "in". This and the rest of the appendix should be corrected.

E. Also with respect to Appendix D, the authors should also explain why the same modified forms appear multiple times in the same list of modifications -- e.g., "not only because (51), not only because (6), not only because (6)"; also "in conclusion (305), in conclusion (269), in conclusion (164)", etc.

F. Figure 2 needs more explanation: There is a drop in both "change flow" and "sense flow" for N=4, which rises again for larger values of N.  This is not explained or even mentioned: all the authors to is claim that a combination of N=1,2 and 3 gives the best result. Is this an artifact? If so, of what? If not, can the authors explain it or even hypothesize an explanation?


**Reasons To Accept:**

While most recent work on discourse relations has focussed on recognizing the sense of implicit relations and doing so for individual tokens, the current paper focusses on problems involving two types of explicit (albeit ambiguous) signals of discourse relations -- explicit connectives and alternative lexicalizations.

**Reasons To Reject:**

While there are problems with the paper, I wouldn't consider any of them to be "reasons to reject".

**Reproducibility:**

4: Could mostly reproduce the results, but there may be some variation because of sample variance or minor variations in their interpretation of the protocol or method.

**Reviewer Confidence:**

4: Quite sure. I tried to check the important points carefully. It's unlikely, though conceivable, that I missed something that should affect my ratings.

---

> ### Author Rebuttal · Authors · 2023-08-28
>
> We thank the reviewer for these extensive and constructive comments.
>
> The questions revealed some lack of clarity in the text, which we would like to resolve:
>
> (A) This was misleading, as we here refer to the predicted signals used for the statistical evaluations (§4.2). We do not use the original annotations for comparison reasons (prediction errors and slightly different senses).
>
> (B) For each of the 10 iterations (§4.1), we use a different split for train/dev/test. However, we make sure that each model setting uses the exact same splitting in its i-th iteration.
>
> (C) We will reformulate the claim and make clear that it does not generalize to coarse senses, as the results are mixed, but for fine senses, the averaged results are all better, although we are missing tests to prove significance. This is mainly because our work is mostly interested in fine senses, while coarse senses are used more as an auxiliary task. For the mixed results on coarse senses, we hypothesize the model to focus on the easier task (classifying 4 types) rather than the more complex fine-sense classification, thus resulting in better coarse-sense scores. However, we can not provide strong evidence for this assumption.
>
> (D) The headline of the appendix is misleading and will be clarified. In particular, the list summarizes the predictions of our connective extraction model, and the list is analogously built to the PDTB3 manual (see below). We will describe the process of creating the list in more detail.
>
> (E) Thanks for pointing out his problem. We missed aggregating the individual results per corpus, thus leading to seemingly duplicate entries. We will correct this.
>
> (F) For our experiments we used n-grams of size up to 4, thus the mentioned drop is for the largest value and thus does not rise further for larger n. Instead, the table then presents results for three combinations of n-grams (1-2, 1-2-3, 1-2-3-4). Regarding the missing interpretation, we hypothesize that a major weak point of our modeling is the missing implicit relations, which account for half of the annotations. When linearizing the discourse senses, we have certain (implicit relation) gaps in the flow. The longer the n-gram we extract, the higher the chances that we end up with patterns with missing discourse information. Those seem more confusing than helpful for the classification model. Therefore, we mentioned accounting for implicit relations as an important step for future work.
>
> Thanks for the missing references. We will mention Biran/McKeown.
> The PDTB Annotation Manual indeed inspired the Table in the appendix, and we will cite it appropriately. In contrast to the table in the manual, our table summarizes automatically extracted explicit discourse signals and further groups them automatically by their connective head, thus alignment problems might occur. We also would like to highlight that these explicit signals in the table represent only a subset of all extracted signals, and they are disjoint from the ones found in the PDTB. We will clarify that in the appendix.

---

### Official Review · Reviewer_n3Y1 · 2023-08-11

**Soundness:** 3

**Excitement:**

3: Ambivalent: It has merits (e.g., it reports state-of-the-art results, the idea is nice), but there are key weaknesses (e.g., it describes incremental work), and it can significantly benefit from another round of revision. However, I won't object to accepting it if my co-reviewers champion it.

**Paper Topic And Main Contributions:**

The paper titled "Discourse Sense Flows: Modelling the Rhetorical Style of Documents across Various Domains" presents three contributions that advance the discourse analysis domain:

Advancements in Signal Extraction Models: The authors developed novel models for signal extraction from discourse relations, specifically concentrating on explicit connectives and alternative lexicalizations. These innovative models are meticulously tailored to classify the nuanced senses of these signals, harnessing the PDTB3 dataset.

Inception of Discourse Sense Flows: The paper proposed "discourse sense flows," an ingenious technique that captures the rhetorical essence of a document by harnessing the sequential arrangement of coherence relations, as characterized by PDTB senses.

Empowering Genre/Domain Discrimination through Discourse Sense Flows: Through comprehensive comparative analyses spanning multifarious corpora, the paper meticulously examined sense flows. The authors leverage these findings to conduct experiments in automatic genre/domain discrimination.

**Reasons To Accept:**

see the main contributions

**Reasons To Reject:**

The paper makes contributions to the field of discourse analysis, yet it also faces significant challenges in its current form.

Lack of Clear Structure: The organization of the paper poses a challenge, as it combines two distinct elements: signal extraction and discourse sense flows. Unfortunately, these components lack a clear connection and do not form a coherent objective or innovative methodology, as suggested by the paper's title.

Methodological Limitations: While the paper's focus on examining discourse relations with lexical signals aligns with PDTB-style discourse analysis, the proposed methods seem to yield only modest enhancements in efficiency. Additionally, the method of discourse sense flows fails to effectively tackle challenges associated with Rhetorical Structure Theory (RST) parsing problems, leaving potential avenues unexplored.

Untapped Potential in Parsing and Analysis: Although the idea of adapting discourse sense flow extraction to RST-style parsing is intriguing, the paper falls short in fully exploring this promising direction. The omission of an in-depth exploration of this concept creates a significant gap in the paper's coverage. Moreover, the absence of a robust comparative analysis against existing methods, such as other RST parsing techniques, hampers the paper's ability to underscore the uniqueness and advantages of its proposed discourse sense flows.

Lack of Comparative Framework: The paper does not provide a thorough comparison between its innovative discourse sense flows approach and established methodologies for rhetorical style modeling and genre/domain discrimination. This absence of comparative benchmarking diminishes the paper's ability to effectively highlight the distinctive contributions it brings to the field.

In conclusion, the paper introduces novel methodologies aimed at identifying and representing discourse sense, with a focus on capturing the inherent rhetorical style of documents through the concept of discourse sense flows. This approach holds promise for facilitating genre/domain discrimination. However, the paper's significance is overshadowed by identified shortcomings, spanning issues related to structural organization, methodological limitations, unexplored avenues, and unverified applications. As it stands, the paper falls short of meeting the rigorous standards expected for acceptance at the EMNLP conference.

**Reproducibility:**

3: Could reproduce the results with some difficulty. The settings of parameters are underspecified or subjectively determined; the training/evaluation data are not widely available.

**Reviewer Confidence:**

3: Pretty sure, but there's a chance I missed something. Although I have a good feel for this area in general, I did not carefully check the paper's details, e.g., the math, experimental design, or novelty.

---

> ### Author Rebuttal · Authors · 2023-08-28
>
> We thank the reviewer for the thoughtful comments. We'd like to respond to the four reasons to reject:
>
> 1) Connection between the two parts of the paper: Shallow discourse parsing is commonly seen as a series of distinct subtasks, one of them being the identification of senses of signals. We focus on this particular subtask and broaden its perspective by incorporating not only connectives but also „alternative lexicalization“, which so far has received very little attention. Then, based on this more comprehensive operationalization of the signal identification step, we show an application (sense flows, genre differences) for this particular subtask, which (i) can be done without the complete shallow discourse parsing machinery, and (ii) yields interesting results. - But we acknowledge that this link between the two parts could have been formulated more clearly and will do so in a revision.
>
> 2) Merely modest enhancements of signal identification: We strongly believe that coherence relations that are signaled not by connectives but by other lexical means need to receive much more attention in the community if SDP as a whole is to improve (because connectives—at least in English—are basically exploited completely by now). The reported improvement appears marginal, but only as the high number of explicit connectives obscures the small number of alternative lexicalizations. Thus, the results of the latter have less weight on the changes in the final scores. A more precise comparison cannot be made because the referenced work did not distinguish these kinds of signals.
>
> 3) Unexplored avenues for RST parsing: We think this is definitely a promising area for future work but beyond the scope of this paper, which for now concentrates on SDP/PDTB. Given the space limits, we don’t think that another paradigm for coherence modeling and parsing can be addressed. Actually, there may be a misunderstanding when the reviewer asks for a „comparative analysis against existing methods such as other RST parsing techniques“: We do not intend to compete with RST parsing; our goal is to enhance the signal identification performance only, and then apply that PDTB-style subtask, as sketched above.
>
> 4) No thorough comparison between discourse sense flows and established methods for rhetorical style modeling: Yes, we acknowledge this comparison needs to be strengthened. In the submission, it was a matter of space, but if given another page for a final version, we will definitely expand our comparison to related work in this regard.

---

### Meta-Review · Area_Chair_25KS · 2023-09-08

**Recommendation:** 3

**Metareview:**

The paper presents models to detect explicit discourse connectives and alternative lexicalisations, as well as to classify their senses (based on PDTB3). The models are applied on texts of various genres and domains. The authors found out that n-gram patterns are stronger predictors than unigram ones.
The paper has several cons: (1) working not only on explicit connectives, but also alternative lexicalisations, which so far has received very little attention;  (2) using sense flows to identify rhetorical patterns of different genres and domains is novel; (3) using various levels of PDTB sense hierarchy; (4) in general a well-structured and a well-described paper enabling the replication of this work.
But it has also some disadvantages: the minor ones include some flaws (e.g. missing references or typos). The major ones include: (1) a lack of clarity in the subtasks or their connection; (2) comparison with the baseline and (3) a terminological confusion in terms of rhetorical analysis.
However, the authors actively participated in the discussion and their rebuttal sounds convincing.r

---

### Decision · Program_Chairs · 2023-10-07

**Decision:**

Accept-Findings

**Comment:**

The paper presents models to detect explicit discourse connectives and alternative lexicalisations, as well as to classify their senses (based on PDTB3). The models are applied on texts of various genres and domains. The authors found out that n-gram patterns are stronger predictors than unigram ones.
The paper has several cons: (1) working not only on explicit connectives, but also alternative lexicalisations, which so far has received very little attention;  (2) using sense flows to identify rhetorical patterns of different genres and domains is novel; (3) using various levels of PDTB sense hierarchy; (4) in general a well-structured and a well-described paper enabling the replication of this work.
But it has also some disadvantages: the minor ones include some flaws (e.g. missing references or typos). The major ones include: (1) a lack of clarity in the subtasks or their connection; (2) comparison with the baseline and (3) a terminological confusion in terms of rhetorical analysis.
However, the authors actively participated in the discussion and their rebuttal sounds convincing.r